# Analysis of Raffinose Synthase Gene Family in Bread Wheat and Identification of Drought Resistance and Salt Tolerance Function of *TaRS15-3B*

**DOI:** 10.3390/ijms241311185

**Published:** 2023-07-06

**Authors:** Jiagui Guo, Yan Yang, Tingting Wang, Yizhen Wang, Xin Zhang, Donghong Min, Xiaohong Zhang

**Affiliations:** 1State Key Laboratory of Crop Stress Biology for Arid Areas, College of Life Sciences, Northwest A&F University, Xianyang 712100, China; gjg1215225@126.com (J.G.); 18398243113@163.com (T.W.); 17791018002@163.com (Y.W.); 18783407676@163.com (X.Z.); 2State Key Laboratory of Crop Stress Biology for Arid Areas, College of Agronomy, Northwest A&F University, Xianyang 712100, China; y15765986512@163.com

**Keywords:** bread wheat, raffinose synthase, *TaRS15-3B*, drought tolerance, salt tolerance

## Abstract

Raffinose synthase (RS) plays a crucial role in plant growth and development, as well as in responses to biotic stresses and abiotic stresses, yet few studies have been conducted on its role in bread wheat. Therefore, in this study we screened and identified a family of bread wheat raffinose synthase genes based on bread wheat genome information and analyzed their physicochemical properties, phylogenetic evolutionary relationships, conserved structural domains, promoter cis-acting elements, and expression patterns. The BSMV-induced silencing of *TaRS15-3B* resulted in the bread wheat seedlings being susceptible to drought and salt stress and reduced the expression levels of stress-related and ROS-scavenging genes in bread wheat plants. This further affected the ability of bread wheat to cope with drought and salt stress. In conclusion, this study revealed that the RS gene family in bread wheat plays an important role in plant response to abiotic stresses and that the *TaRS15-3B* gene can improve the tolerance of transgenic bread wheat to drought and salt stresses, provide directions for the study of other RS gene families in bread wheat, and supply candidate genes for use in molecular breeding of bread wheat for stress resistance.

## 1. Introduction

Bread wheat (*Triticum aestivum* L.), as one of the first plants domesticated by human beings, is widely used in human food and livestock feed species [1]. With the rapid growth of the world population, the safe production of bread wheat is facing enormous challenges [2,3]. Due to the adverse effects of environmental pressures such as drought, high salinity, high and low temperatures, and the attack of pathogens and pests, its yield and quality suffer significant losses [4]. These abiotic stresses damage the development and growth of plants and cause considerable losses to agricultural productivity worldwide [5,6]. Abiotic stress can lead to changes in the physiological morphology, molecular and biochemical properties of plants, and conventional metabolism of plants, which greatly inhibits the genetic potential of plants and may account for the sharp decline in crop yield [7].

Due to global climate change and human activities, extreme weather such as drought is occurring more frequently, causing serious negative effects on crop production, and food security production is facing great challenges [8]. The impact of drought on crop production depends on a wide array of factors, including the intensity and duration of the drought period, and the developmental stage of plants at which deficiency occurs [9,10]. In the initial stage of drought stress, plants usually show poor growth, reduced leaf water content, low turgor pressure [11], and a low transpiration rate [12]. Further drought stress leads to damage to cellular functions such as protein synthesis, cell membrane activity, and nitrogen assimilation [13]. During the germination period of bread wheat seeds, if the seeds cannot absorb enough water, the germination rate of plants is reduced, and ultimately the total number of plants per unit area is reduced [14]. At the seedling stage of bread wheat, drought stress leads to an abnormal expression of genes, and ultimately leads to the decline of seedling vitality [15]. Drought can seriously affect the process of meristematic cell division and cell elongation, resulting in delayed plant growth [16]. In addition, drought stress significantly negatively affects various physiological, biochemical, and agronomic traits of plants [17]. The reduction in leaf number and total leaf area caused by drought will further reduce the photosynthetic activity of plants. When plants are exposed to moderate-to-severe drought, this effect is more prominent in terms of plant height, leaf surface area, and biomass [18] because a water deficit in the early stage of nutrition inhibits cell expansion, resulting in the reduction in leaf area, internode elongation, and plant height [19]. At the same time, all determinants of crop yield and quality are affected by drought stress. Crop yield and quality are complex processes regulated by multiple genes. Drought reduces the size and number of flowers, pollen, and pollen vitality [20], while the quantity and quality of pollen affects the overall pollination [21], and pollination is directly related to the yield of fruits and seeds. While in the breeding stage, water loss not only reduces the number of panicles per plant or the number of grains per panicle but also reduces the weight of grains [22]. Under moderate drought conditions, grain production decreased by 14%. Conversely, under severe drought conditions, grain production decreased by 40% [18].

In addition, salt stress is also an important abiotic constraint on plant distribution, growth, and yield in arid and semi-arid regions [23]. More than 6% of the world’s land is considered to be saline–alkali land [24], and 20% of irrigated land is saline–alkali land, a fact that causes about USD 27.3 billion in agricultural losses annually [25]. With the increase in world population and the change in climate and environment, the effective use of marginal saline–alkali land constitutes an important way to meet the demand for crop productivity [26]. Plants alter cell properties by sensing osmotic changes and ionic signals and transmitting them to the interior of cells [27]. The initial stage of salt stress causes various physiological changes, such as cell membrane disruption and nutrient imbalance, as well as reductions in the ability to detoxify reactive oxygen species, in photosynthetic activity and in stomatal aperture [28]. A large amount of Na^+^ a and Cl^−^ ions accumulate in plant tissues exposed to soil with a high salt concentration. The excessive intake of ions into cells leads to a serious ion imbalance, causing serious physiological disorders [29]. A high Na^+^ concentration inhibits the absorption of K^+^, and K^+^ is a necessary element for growth and development. The decrease in K^+^ leads to the decline of cell viability, and may even lead to cell death [30]. In addition, the generation of ROS is enhanced under salt stress [31]. Excessive accumulation of ROS induced by salt stress can lead to oxidative damage of various cellular components such as proteins, lipids, and DNA, and interrupt important cellular functions of plants [32].

Plants have evolved a variety of mechanisms to cope with suboptimal environments via methods such as triggering a series of signal transduction reactions and accumulating compatible metabolites. The accumulation of raffinose family oligosaccharides (RFOs) is a particularly important part of this process [33]. Galactitol and raffinose are ubiquitous in plants and have been shown to play an important role in seed desiccation tolerance/seed storage [34,35]. In recent years, an increasing number of studies has shown that galactitol and RFOs are related to various abiotic stress responses of plants and demonstrated that the level of RFOs in a variety of plants can be used as an indicator of plant stress tolerance [36,37].

Raffinose plays an important role in plant development. The level of RFOs in *RS4/5* double-knockout lines in Arabidopsis thaliana is reduced, resulting in a 5-day delay in germination in the dark, and the expression of germination inhibitors is up-regulated [38]. When the decomposition of RFOs in pea seeds is blocked, the seed germination rate is significantly reduced [39]. In addition, raffinose also plays an important role in plant resistance to abiotic stress. Firstly, the accumulation of raffinose is related to the drying resistance and high-temperature drying resistance of cereal seeds. In contrast, the inability of raffinose to accumulate effectively leads to the loss of desiccation tolerance during corn seed germination. The overexpression of *RS5* causes *Arabidopsis thaliana* to show a stronger desiccation tolerance [40]. Under dry conditions, high RFOs levels may be required to maintain stable reducing monosaccharide levels in order to confer desiccation tolerance to seeds before germination [35]. In addition to playing an important role in drought tolerance, raffinose also maintains seed vitality and the longevity of plants [34,36]. In hybrid rice seeds, raffinose content is positively correlated with seed germination rate under natural ageing conditions [41,42], while galactose content is negatively correlated with seed germination rate under natural ageing and artificial ageing conditions [43]. In maize seeds, low levels of raffinose lead to lower seed vigor [44]. Similarly, when the expression level of the *ZmRS* gene is reduced, the content of Raffinose in plant seeds is reduced, leading to the shortening of seed life [36]. Raffinose is the only RFOs accumulated in maize seeds, while the seeds of the *zmrs* mutant lacking raffinose show a significant decline in viability even if they survive drying [34]. Meanwhile, maize *zmrs* mutants lack raffinose and show decreased drought resistance, while Arabidopsis *ZmRS*-overexpressing plants show enhanced tolerance to drought stress. The enhanced drought resistance to the overexpression of *zmrs* is due to the increase in inositol level after galactitol hydrolysis, and the increase in inositol to raffinose ratio positively regulates the response of plants to drought stress [40]. When *Thellungiella* is subjected to salt, drought, or cold stress, its raffinose level increases, and the ratio of raffinose to sucrose also increases [45].

While raffinose synthase (RS) is the key enzyme catalyzing raffinose synthesis, the expression level of the *RS* gene is induced by plant seed growth and development and stress, and *AtRS5* (*At5g40390*) is involved in raffinose accumulation in leaves under abiotic stress conditions in Arabidopsis [46]. *GmRS2A* and *GmRS2B* are highly expressed in the middle stage of soybean seed maturation, and raffinose accumulation is also significantly increased [47]. In maize species, the seed viability of the deletion mutant of *ZmRS* is reduced [34], while the overexpression of *AtRS5* can significantly improve the germination rate of Arabidopsis seeds and drought resistance of seedlings [35]. Under conditions of cold stress, *RS* gene expression is up-regulated in rice, corn, and melon [48,49,50]. However, under drought conditions, high temperature and high salt stress, *RS* gene expression levels are up-regulated to varying degrees [40,51]. In addition, some transcription factors also participate in the regulation of *RS* expression and then regulate raffinose biosynthesis to affect resistance. The overexpression of the Arabidopsis heat shock transcription factor *AtHsf2* can induce the expression of *AtRS2* and the increase in raffinose content, resulting in transgenic plants with stronger stress resistance [52]. While the overexpression of *BnHsf4a* in Arabidopsis can also induce the expression of *AtRS2* [46], *ZmHsf2* is a corn heat shock transcription factor that can induce the expression of *RS*, thereby improving the heat tolerance of corn [53]. The overexpression of *OsWRKY11* in rice can increase the expression of *OsRS* and raffinose content, thereby improving drought resistance [54]. Other studies have shown that transcription factors such as *CBF* and *ERF* can also improve plant stress resistance by regulating the expression level of *RS* [36,55].

The *RS* gene family has been identified in *Arabidopsis thaliana* [46], *Zea mays* L. [56], *lens culinaris Medik* [57], *Gossypium* spp. [58], and *Populus* L [59]. Two motifs (KxD and Rxxxd) are found in RS-related proteins in the conserved domain of the glycoside hydrolase family. The aspartic acid residue in the KxD motif is considered to be a catalytic nucleophile, while the Asp residue in Rxxxd motif is considered to be a catalytic acid/base [60]. However, *RS* gene family members in bread wheat have not been systematically screened or studied. Therefore, this study uses bioinformatics methods to screen and identify the *RS* family in bread wheat based on bread wheat genome information and analyzes its physical and chemical properties, phylogenetic and evolutionary relationships, conserved domains, promoter cis-acting elements, and expression patterns. On this basis, the drought and salt tolerance function of candidate gene *TaRS15-3B* is evaluated. This provides a reference for the study of other members of the *RS* gene family in bread wheat.

## 2. Results

### 2.1. Analysis of RS Gene Family in Bread Wheat

#### 2.1.1. Screening and Phylogenetic Analysis

A total of 34 genes encoding raffinose synthase were screened from the bread wheat genome, and these genes were renamed according to their positions on the bread wheat chromosome [61]. Bread wheat *RS* gene family members all possess a Raffinose_ Syn (PF05691)-conserved domain, but their physicochemical properties are very different. For details, see Appendix A. The molecular weights of these 34 TaRS proteins ranged from 10.80 KDa (TaRS30-7A) to 120.85 KDa (TaRS11-2B), with an average molecular weight of 59.37 KDa. In addition, basic protein accounted for only 38.24% (13/34) of the TaRS proteins. A total of 73.52% (25/34) of TaRS proteins have been found to possess an instability index of less than 40, indicating that most of them are stable proteins. The fat group index of TaRS was greater than 66, which also indicated that TaRS protein had certain thermal stability. The hydrophilic indexes of TaRS proteins were all negative, indicating that they were all hydrophilic proteins. TaRS protein, which may fulfil a variety of biological functions, is distributed in various organelles.

To explore the phylogenetic relationship of RS protein members in bread wheat, we constructed a phylogenetic tree using 34 selected TaRS protein sequences of bread wheat (Figure 1). In addition, a phylogenetic tree was constructed using 164 protein sequences of RS protein family members found in monocots and dicots (Figure 2). We divided the RS proteins in plants into four subfamilies according to the results of protein comparisons and previous research results obtained for Arabidopsis and cotton [58]: RS I, RS II, RS III, and RS IV.

The phylogenetic tree results showed that the members of the four subfamilies of RS proteins were distributed in seven species, *Triticum aestivum*, *Aegilops tauschii Coss*, *Oryza sativa Japonica*, *Zea mays*, *Arabidopsis thaliana*, *Glycine.max*, *Gossypium raimondii*, *Solanum tuberosum* L., and *Solanum lycopersicum* L. (Appendix A), suggesting that the RS protein family was conserved during evolution. In addition, its phylogenetic relationship with the monocotyledonous plant *Aegilops tauschii Coss* was closer, and TaRS potentially originated from the allopolyploidization of bread wheat. At the same time, the proportion of RS protein in the number distribution of RS I, RS II, RS III, and RS IV subfamilies in bread wheat was 44.12%, 8.82%, 14.70%, and 32.38%, respectively. RS I and RS IV in the RS gene family in bread wheat accounted for 76.5% of the number of members in the family. In the same way, in other species, the members of the RS family were the same, suggesting that RS I and RS IV families expanded in RS-like proteins (Appendix A).

#### 2.1.2. Motif Analysis and Gene Structure Analysis

The results of conserved motifs in bread wheat TaRS proteins predicted by MEME (Figure 3 and Appendix A) showed that the types and numbers of protein motifs within each subfamily of TaRS were very similar, but varied greatly among subfamilies. For example, Motif 2, 8, 3, and 9 are well-conserved throughout the TaRS family, occurring in 82.35% (28/34) of the family members and, presumably, the related Motif is important in TaRS protein structure and function. However, motif 10 is unique to the RS IV subfamily and is presumably related to specific protein functions. In addition, TaRS7-1A, TaRS9-2A, TaRS19-4A, TaRS26-5D, TaRS27-5D, and TaRS30-7A only showed Motif 1, and no clear protein translation region and matching protein-conserved motifs were found in the upstream 5000 bp gene fragment, which may be subject to extensive mutations or gene recombination.

The results of the conserved structural domain of the protein, however, were similar to those of the conserved motifs, and the missing fragments TaRS7-1A, TaRS9-2A, TaRS19-4A, TaRS26-5D, TaRS27-5D, and TaRS30-7A also matched Raffinose_syn (PF05691), indicating that Motif 1 may be the structural domain of the signature fragment. Similarly, the gene lengths and exon numbers of the TaRS gene family differed significantly among the subfamilies, but were generally consistent with the results of conserved motifs and conserved structural domains. The differences in protein motifs, conserved structural domains, and gene structures among the subfamilies may explain the functional diversity of the TaRS subfamily proteins.

#### 2.1.3. Chromosome Mapping and Homologous Gene Identification

The *TaRS* gene family was localized to 16 chromosomes in bread wheat (Figure 4, Appendix A). *TaRS* genes were most abundantly distributed on chromosomes 1A and 5D, with four. Conversely, no *TaRS* genes were localized on chromosome group 6, and there were only two chromosomes 4B and 4D. In addition, *TaRS* was unevenly distributed on the chromosomes of the bread wheat genome and showed significant uneven distribution in its four subfamilies, and each chromosome evolved relatively independently, presumably with an independent mechanism of genetic variation in the evolution of the *TaRS* gene.

As a member of a typical heterozygous polyploid evolutionary branch with complex species relationships, a genome size of 17 Gb, and a high proportion of duplicated and replicated genes [62], most bread wheat genes exhibit 1:1:1 homozygosity due to two rounds of polyploidization. In contrast, the proportion of homozygous triplicates (52.94%), loss of one homozygous gene (11.76%), and homozygous-specific duplicates (17.65%) in the bread wheat *RS* gene family is higher than that in the whole-bread wheat genome (Table 1). However, the proportion of orphan genes (Orphans/singletons) in the whole-bread wheat genome was 37.1%, while the proportion of orphan genes in the *TaRS* gene was only 2.94%, meaning that it is reasonable to presume that bread wheat polyploidization was the primary reason for the expansion of the *TaRS* family in bread wheat.

#### 2.1.4. Gene Family Collinearity and Replication Event Analysis

To further explore the evolutionary relationships of the bread wheat *RS* gene family, we conducted an analysis of bread wheat gene family covariation and duplication events (Figure 5). This revealed that 25 *TaRS* genes in the bread wheat genome constituted synonymous regions, forming 20 duplicated gene pairs (Appendix A). Furthermore, 44% (11/25) of the *TaRS* genes were clustered on chromosomes 1 and 3, corresponding to the large number of *TaRS* genes present on them. Meanwhile, 73.53% (25/34) of *TaRS* genes were involved in gene duplication events, whereas all *TaRS* genes constituting the synonymous domain region in bread wheat were derived from WGD/fragmental duplication events and no fragmental duplication events were present, suggesting that WGD/fragmental duplication events are likely to be among the main causes of the expansion in the number of *TaRS* genes, which is consistent with the identification of homologous *TaRS* gene families in bread wheat.

In order to further understand the evolutionary relationships of the *TaRS* genes, a genome-wide covariance analysis was also performed between bread wheat and seven other plants (*Gossypium raimondii*, *Zea mays*, *Oryza sativa Japonica*, *Arabidopsis thaliana*, *Solanum tuberosum* L., *Solanum lycopersicum* L., and *Aegilops tauschii Coss*) (Figure 6). Its results revealed that the *TaRS* gene family harbors many more collinearity blocks in the monocotyledonous species *Aegilops tauschii Coss*, *Oryza sativa Japonica*, and *Zea mays* (Appendix A), suggesting that RS proteins are conserved through species evolution.

In parallel, among nonredundant collinear genes of the *TaRS* gene family for different species (monocot: *Aegilops tauschii Coss*, *Zea mays*, *Oryza sativa Japonica*; dicot: *Arabidopsis thaliana*, *Glycine.max*, and *Solanum lycopersicum* L.) (Figure 7a and Appendix A), eight *TaRS* gene family members showed collinear blocks in all three monocots, whereas only one *TaRS* gene showed collinearity in all three dicots. This is in fact the only member of the *TaRS* gene family that has a colinear block in all six species. In addition, there were significantly more collinearity blocks in the *Aegilops tauschii* than in other species, indicating that the *RS* gene family is evolutionarily conserved among species and that the *TaRS* gene family in bread wheat may have been derived from the polyploidization process of bread wheat, which is in accordance with the above results.

Conversely, there are Ka/Ks values of less than 1 in the isogenic pairs of bread wheat and coarse goat, rice, as well as soybean (Figure 7b), indicating that *TaRS* has evolved through purification and selection. In contrast, only some of the bread wheat homeologs with Leydig have Ka/Ks values greater than 1, presumably resulting from continuous positive selection for excellent traits.

#### 2.1.5. Cis-Acting Element Analysis of Promoter Region

A total of 54 cis-acting elements were identified for the *TaRS* genes (Figure 8), each of which contained multiple TATA box and CAAT box containing common cis-acting elements in promoter and enhancer regions, suggesting a condition for normal transcription.

In addition, among the cis-acting elements screened in the *TaRS* gene (Figure 8), the number of stress-related elements was the highest, accounting for approximately 20% of the elements (554/2782). This was followed by hormone-responsive elements with about 17% (454/2782), and light-responsive elements with about 14% (371/2782), while plant growth responsive elements were the least represented with only 5.9% of the elements (163/2782). Among the hormonal response elements, most were responsive methyl jasmonate (MeJA) response elements (41.85%), as well as abscisic acid (ABA) response elements (33.70%). Conversely, among the elements related to environmental stress, those related to drought and salt stress were dominant and reached 80.87%. These results suggest that the *TaRS* genes may be involved in abiotic stress responses in plants, especially stress-related drought.

In parallel, heat map results of important cis-acting elements commonly found in the promoter of the *TaRS* gene showed (Appendix A) that specific elements associated with environmental stress (MYB and MYC) versus hormones (ABRE, CGTCA-motif, and TGACG-motif) were present in almost every member, indicating that members of the *TaRS* gene family may be involved in plant stress resistance and hormone stress response through these cis-acting elements. In addition, the light-corresponding element (G-box), as well as the auxin-related element (as-1), were also present in most of the members. These elements were speculated to be involved in plant light-related growth and development processes, i.e., in a fashion consistent with the role of *AtRS5* in promoting Arabidopsis seed germination in darkness [38].

#### 2.1.6. Gene Ontology (GO) Analysis and Protein Interaction Network (PPI) Analysis

The predicted protein interaction network for TaRS (Figure 9a) shows that most of the proteins that interact with it are from the α-galactosidase family (*Traes_2AS_BA96EE8B5.2*, *Traes_2DS_7F07924BE.1*, *Traes_1DL_4A0C06C2F.2*) and the glycosyltransferase 8 family (*Traes_2AS_7339C00EB.1*, *Traes_2BS_BC7099C57.1*, *Traes_2BS_E69390845.1*, *Traes_4AS_AF7143BFC.1* and *Traes_4AS_AF7143BFC1.1).* However, due to the large genome of bread wheat and the incomplete information of the annotated genes, most of the proteins do not indicate their major roles.

Therefore, the homologs of TaRS in Arabidopsis were used to predict protein interaction networks, which showed (Figure 9b) that: the RS family proteins were associated with proteins involved in plant metabolism, growth, and differentiation, such as the plant starch mobilization proteins AMY1, AMY2, and AMY3 [63]; the proteins Gols3, 5, 6, and 10 respond to galactinol and RFO synthesis against abiotic stresses [64,65,66]; *HEXO1*, *2*, and *3* are involved in glycoprotein synthesis in plants [67]; galactosidases (*BGAL2*, *17*) are involved in the hydrolysis of the raffinose synthase family of oligosaccharides during seed germination [68]; proteins CWINV4 and 5 are involved in the synthesis and hydrolysis of the cell wall in response to abiotic stresses and sugar metabolism [69,70,71]; and glycosyl hydrolase family related *AT1G62660*, *AT5G11720*, and *AT3G56310*, and other proteins played key roles. The analysis of the protein interaction network suggests that the TaRS gene family is involved directly or indirectly in a variety of plant biological activities through sugar metabolism in plants, particularly in plant responses to abiotic stresses.

In addition, the protein sequence was analyzed for GO annotations to better understand the biological function of *TaRS* (Figure 10, Appendix A). Due to the large number of GO annotations for the *TaRS* gene, we only show the most frequent GO annotation entries, as can be seen in Appendix A. The results show that the *TaRS* gene is involved in various biological processes, such as metabolism (GO:0008152), carbohydrate metabolism (GO:0005975), stress response (GO:0050896), and oligosaccharide metabolism (GO:0009311). In terms of molecular function, it is involved in catalytic (GO:0003824), raffinose α-galactosidase (GO:0052692), glycosyl hydrolase (GO:0016798), and glycosyl hydrolase activity (GO:0016757). Meanwhile, analysis of cellular components indicated that TaRS proteins may be located in intracellular organelles, membrane-bound organelles, and cytoplasm. Additionally, our research indicated that they are involved in the synthesis and breakdown of plant-type cell walls. Based on these results, it is hypothesized that the *TaRS* gene family may be involved in plant responses to abiotic stresses by regulating secondary metabolic processes such as oligosaccharides, which in turn control the production and degradation of the cell wall and seed-stored nutrients. These results are consistent with the above analysis of cis-acting elements.

#### 2.1.7. Analysis of Expression Patterns of TaRS Genes

To understand the expression of *TaRS* genes in different tissues and growth stages of bread wheat, we downloaded RNA-seq data from the bread wheat expression database for the roots, stems, leaves, and spikes of Chinese spring bread wheat seedlings at the vegetative growth stages. The results of the tissue-specific expression heat map showed (Figure 11) that 23.5% (8/34) of the TaRS genes exhibited high levels of expression at almost all developmental stages (log_2_(tpm + 1) ≥ 1), indicating that they are essential for the entire developmental stage of bread wheat. In contrast, 17.6% (6/34) of *TaRS* genes showed very low or no expression (log_2_(tpm + 1) < 1) at all in the developmental stages, which is because they undergo a functional divergence and functional redundancy. At the same time, *TaRS* gene expression levels were concentrated in the leaf/stem region of bread wheat; however, its expression was significantly lower in the spike than in other tissues, while the *TaRS* IV subfamily had higher expression levels in the seed as well as in the root. Notably, *TaRS13-3A*, *TaRS15-3B*, and *TaRS17-3D* in the *TaRS* III subfamily were only expressed during specific periods of leaf development and were presumably essential for the morphological development of bread wheat leaves.

The results of the heat map of *TaRS* gene expression differences under adversity stress showed (Figure 12) that, in terms of the overall trend, *TaRS* gene expression levels of *TaRS* subfamilies I, III, and IV were differentially up-regulated under osmotic, drought, and high-temperature stresses. This was in contrast to the results obtained with *TaRS* subfamily II, which showed slight down-regulation under various abiotic stresses. In addition, *TaRS3-1A*, *TaRS5-1B*, *TaRS6-1D*, *TaRS7-1D*, *TaRS720-5A*, *TaRS23-5B*, *TaRS25-5D*, and *TaRS26-5D* were only up-regulated in chromosome groups 1 and 5 in response to low-temperature stress. In contrast, *TaRS13-3A*, *TaRS15-3B*, and *TaRS17-3D* on chromosome 3 of the TaRS subfamily III showed significant up-regulation under drought and high-temperature stress. Referring to the expression of *TaRS* genes in various tissues, these three members were not expressed throughout the developmental stages of the tissue, but showed significant elevation under conditions of drought and high-temperature stress, which presumably occurs through their involvement in plant leaf development or tissue regulation in response to drought and high-temperature stress.

To verify the previous inference, we selected four genes, *TaRS3-1A*, *TaRS10-2A*, *TaRS15-3B*, and *TaRS18-3D*, from the subfamilies of *TaRS* and genes containing more cis-acting elements related to plant stress, and explored their expression patterns by qRT-PCR under drought, NaCl, and ABA conditions. The expression patterns of these four genes under drought, NaCl, ABA, and MeJA stresses were investigated by qRT-PCR (Figure 13). We found that the expression levels of all four genes changed under stress treatment, but that the extent and pattern of change varied.

Under drought stress, the expression levels of *TaRS3-1A*, *TaRS10-2A*, *TaRS15-3B*, and *TaRS18-3D* all showed a trend of up-regulation followed by down-regulation, but *TaRS18-3D* was less obviously up-regulated than the other three genes and peaked at 12 h, while the other three genes reached their highest values at 24 h. The overall trend of expression under ABA hormone treatment showed an increase followed by a decrease, but peaked at 6 h, 12 h, 24 h, and 6 h. The expression of *TaRS10-2A* and *TaRS15-3B* showed significant up-regulation after MeJA hormone treatment, with the highest expression levels at 12 h and 24 h, respectively. Similarly, the expression levels of *TaRS10-2A*, *TaRS15-3B*, and *TaRS18-3D* showed an explosive increase at 12 h after NaCI treatment and then started to decrease before returning to the initial state. The above results showed that the expression patterns of the four selected TaRS subfamilies, namely *TaRS3-1A*, *TaRS10-2A*, *TaRS15-3B*, and *TaRS18-3D*, differed greatly under drought, NaCl, ABA, and MeJA stresses. These findings were consistent with the results of previous protein-conserved motif, cis-acting progenitor, and GO enrichment analysis.

#### 2.1.8. Analysis of Expression Patterns and Characterization of TaRS15-3B

In the previous section, we found that three homologs of the *TaRS* III subfamily, namely, *TaRS13-3A*, *TaRS15-3B*, and *TaRS17-3D*, exhibited specific expression forms that were highly consistent across developmental periods and under abiotic stresses, especially under drought and high-temperature stresses, and were significantly up-regulated, in addition to sharing similar conserved motifs. The above results suggest that these three genes may exercise a similar role in the development of the gene. These results suggest that the three genes are regulated by upstream cis-acting elements and thus perform the same function in response to drought stress in bread wheat.

The results of multiple sequence alignment tests of TaRS13-3A, TaRS15-3B, and TaRS17-3D with RS proteins extracted from *Zea mays*, *Aegilops tauschii Coss*, and *Glycine.max* (Figure 14) revealed that each contained two conserved modules, KxD and RxxxD, which are considered to be the active centers of cottonseed sugar synthase [72]. In accordance with the results of the previous chapter, these proteins clustered to the same branch of the evolutionary tree as the RS proteins in *Aegilops tauschii Coss*, *Oryza sativa Japonica*, *Zea mays*, while TaRS13-3A, TaRS15-3B, and TaRS17-3D proteins showed 98.38% similarity (Appendix A), with 88.99% nucleic acid sequence similarity.

The expression patterns of *TaRS13-3A*, *TaRS15-3B*, and *TaRS17-3D* were examined in different tissues of bread wheat (root, stem, leave, glume, stamen, pistil, and seed) (Figure 15). The results showed that the expression of *TaRS15-3B* was relatively higher in leaves and seeds compared with *TaRS13-3A* and *TaRS17-3D*. Meanwhile, *TaRS15-3B* expression levels were significantly up-regulated under drought, salt, ABA, and MeJA treatments in the previous chapter, so the *TaRS15-3B* gene was selected for the next functional analysis.

To further investigate the role of the *TaRS15-3B* gene in bread wheat growth and development and abiotic stresses, further information was analyzed. We discovered that TaRS15-3B has a molecular formula of C_3804_H_5863_N_1037_O_1110_S_35,_ a total atomic number of 11,849, a molecular weight of 85.01 kDa, and a theoretical isoelectric point pI of 5.36, making it an acidic protein. Its aliphatic amino acid index is 84.51, which is thermally stable; the average hydrophobicity is −0.074, indicating that the protein is hydrophilic; and its instability coefficient is 33.92 < 40, indicating that it is a stable protein. In addition, the secondary structure of the TaRS15-3B protein was predicted and was found to have a high proportion of Alpha Helix.

The promoter region of *TaRS15-3B* contains a variety of cis-acting elements (Appendix A), such as ABRE, CGTCA-motif, O2-site, P-box, TGACG-motif, TGA-element, and other hormone-responsive cis-acting elements; growth- and development-related cis-acting elements such as motif I (root-specific expression element) and MSA-like (cis-acting element involved in cell cycle regulation); and stress-related cis-acting elements such as ARE, DRE core, GC-motif, MBS, MBSI, MYB, MYB, MYC, and STRE. Of these, the abscisic acid response element ABRE was the most abundant with 17 copies, and TaRS15-3B was likely to be further involved in plant responses to abiotic stresses through the plant ABA signaling pathway. This is consistent with its elevated expression level in response to ABA treatment.

### 2.2. Functional Characterization of the TaRS15-3B Gene for Drought and Salt Tolerance 

#### 2.2.1. Acquisition and Characterization of Overexpressed TaRS15-3B Bread Wheat

To investigate the role of *TaRS15-3B* in the response of bread wheat to abiotic stress, the *TaRS15-3B* gene was constructed into the pWMB003 vector and the T3 generation of bread wheat overexpressing *TaRS15-3B* was obtained via a gene gun bombardment of the healing tissues of the bread wheat variety KN199. The transgenic bread wheat line with high expression were identified by PCR and qRT-PCR for the next functional identification of the transgenic bread wheat (Appendix A).

#### 2.2.2. Evaluation of Salt and Drought Tolerance in TaRS15-3B Overexpression Lines of Bread Wheat during Germination Stage

To investigate the role of the *TaRS15-3B* gene in the germination of bread wheat under drought and salt stress, intact and uniformly sized seeds of the control and the overexpression line of the *TaRS15-3B* gene were selected and simulated with 20% PEG 6000 and 150 mM NaCl to determine the number of seeds germinated on the third and seventh days. The results showed that (Figure 16) under normal moisture conditions there was no significant difference in the germination rate between the wild-type and the *TaRS15-3B*-overexpressing strain, but the transgenic strain grew significantly better than the control; while under 20% PEG and 150 mM NaCl, the germination potential and germination rate of the seeds were significantly reduced, but the *TaRS15-3B*-overexpressing strain was significantly better than the control. The above results indicate that the *TaRS15-3B* gene line showed a higher tolerance to drought as well as salt stress during the germination period of bread wheat.

#### 2.2.3. Identification of Drought and Salt Resistance Function in Seedlings of Overexpressing TaRS15-3B Bread Wheat

##### Identification of Drought Resistance Function in Seedlings of Overexpressing TaRS15-3B Bread Wheat

To investigate the effect of the *TaRS15-3B* gene on bread wheat plants under drought stress, bread wheat seedlings of similar size were selected from overexpressing *TaRS15-3B* and wild-type gene lines at the second leaf stage and subjected to drought stress using hydroponics and soil culture, respectively. The results of the simulated drought stress treatment with 20% PEG6000 in the hydroponic method (Figure 17) showed that after 5 days of drought stress, the bread wheat seedlings started to show dehydration symptoms, with the leaves losing their green color and curling and wilting; after 8 days of drought stress, the bread wheat seedlings basically showed severe dryness. However, compared with the control, the *TaRS15-3B* overexpression strain showed significant drought tolerance, with the leaves wilting. However, compared with the control, the *TaRS15-3B* overexpression strain showed significant drought tolerance, with low leaf wilting, and was able to remain upright after five days of drought stress. After 3 days of rehydration with water instead of PEG solution, the overexpressed *TaRS15-3B* strain started to turn green and resume growth, while the seedlings of the control died and could not continue to grow.

Seedlings of uniform growth and in good condition were selected and transplanted into nutrient soil. After one week of growth in the incubator, they were subjected to natural drought stress. The results of the drought stress on the wild-type and overexpressing *TaRS15-3B* line (Figure 17b) showed that after 12 days of drought stress, the bread wheat seedlings showed severe chlorosis and leaf drying, with the wild-type wilting and drying to a more pronounced extent than the transgenic strain. After 7 days of rehydration, the TaRS15-3B overexpression line basically regained growth, while the wild-type strain was almost completely dead and could not grow further. The *TaRS15-3B* overexpression strain was more resilient and had a significantly higher survival rate than the wild-type strain. At the same time, we explored the physiological mechanisms of the wild-type and overexpression *TaRS15-3B* lines before and after drought stress (Figure 17c–g), and found that after drought stress, compared with the wild-type, the MDA content of the overexpression TaRS15-3B lines showed a highly significant decrease, while the proline content, POD activity, and CAT activity all showed a highly significant increase.

##### Identification of Salt Tolerance in Seedlings of Overexpressing *TaRS15-3B* Bread Wheat

After treatment with 200 mM NaCl solution to simulate salt stress (Figure 18a), we found that bread wheat seedlings began to show leaf curl and yellowing of leaf tips only after 7 days of treatment, and bread wheat growth was inhibited; after 12 days of treatment, the leaves became dehydrated and curled and wilted, and widespread collapse occurred. However, compared with the wild-type strain, the *TaRS15-3B* overexpression strain showed significant salt tolerance, with less leaf wilting, and the leaves curled and wilted but remained upright after 12 days of salt stress. Similar to drought stress, there were no significant differences in the physiological indicators (MDA content, PRO content, POD activity, CAT activity, and SOD activity) of bread wheat seedlings before salt stress treatment (Figure 18b–f). After the application of salt stress, there was a highly significant decrease in MDA content and a highly significant increase in proline content and SOD activity, in addition to a significant increase in POD activity in the *TaRS15-3B* overexpression strain compared with the wild-type strain.

The above results show that, in the seedling stage of wheat, overexpressing the *TaRS15-3B* gene resulted in improved tolerance to drought and salt stress. The exploration of its underlying physiological mechanism shows that it may enhance plant drought resistance and salt tolerance by participating in the accumulation of free proline and enhancing ROS-scavenging enzyme activity.

#### 2.2.4. Evaluation of Drought Tolerance in TaRS15-3B Overexpression Lines of Bread Wheat during the Middle and Late Stages of Development

Bread wheat is one of the most important food crops in the world. As such, the late growth stage of bread wheat that determines its yield and quality is very important. To study the effect of the *TaRS15-3B* gene on the later stage of bread wheat growth and its development under drought stress, the seedlings of a similar size and growth state were selected and moved into pots. These were buried in the soil to maintain ground temperature, normal water conditions (WW), and drought stress conditions (WD), and we observed and sampled the plants at the jointing stage, booting stage, heading stage, and filling stage of bread wheat. The results showed (Figure 19) that under drought stress, bread wheat plants showed obvious dehydration symptoms, leaf curling and chlorosis, and plant dwarfing. Additionally, the proportion of effective tillers decreased.

In the filling stage (15 days after flowering), the plant height, effective tillers, flag leaf length, flag leaf width, and spikelets of overexpressed *TaRS15-3B* and wild-type bread wheat were investigated under two water conditions. The results showed that (Figure 20, Table 2) under drought stress, the flag leaf length, flag leaf width, plant height, and effective tillers of each line were extremely significantly decreased, and the spikelets were significantly decreased. Compared with the overexpressing *TaRS15-3B* gene line, under drought stress, the wild-type wheat strain showed more obvious dehydration symptoms. Initially, the plant height and spikelet number decreased significantly, and the flag leaf width decreased insignificantly, but the flag leaf length and effective tiller number also decreased significantly, which further led to a reduction in flag leaf area. These occurrences affected the photosynthesis of bread wheat at the filling stage, thus causing a decline in bread wheat yield. The above results showed that the bread wheat strain overexpressing the *TaRS15-3B* gene in the middle and late stages of growth and development showed obvious resistance to drought stress.

At the same time, the expression level of the *TaRS15-3B* gene was detected at the jointing stage, booting stage, heading stage, and grain-filling stage of bread wheat. The results showed that (Figure 21) the expression level of the *TaRS15-3B* gene showed a gradual upward trend in the middle and late stages of bread wheat growth and development. In addition, compared with normal water conditions, drought stress led to the significant upregulation of *TaRS15-3B* expression in various periods, especially in the heading stage and filling stage. As the key period of bread wheat ear development, the heading and grain filling period is the decisive period that affects the yield and quality of bread wheat. In addition, previous studies have shown that the *TaRS15-3B* gene is highly expressed in bread wheat seeds. The increase in *TaRS15-3B* expression during seed development may further affect seed vitality, thus showing a higher germination ability under drought and salt stress. This is consistent with the results of 2.2.2.

### 2.3. BSMV-Induced Gene-Silencing of TaRS15-3B and the Identification of Drought and Salt Tolerance Function after Silencing

#### 2.3.1. BSMV-Induced Gene-Silencing of TaRS15-3B

To further clarify the role of *TaRS15-3B* in the response to drought and salt stress in bread wheat seedlings, *TaRS15-3B* gene silencing was induced by the Barley streak mosaic virus (BSMV). The *TaRS15-3B* gene silencing system was constructed by obtaining silencing fragments and performing BSMV: γ-*TaRS15* vector construction, vector linearization, and in vitro transcription (Appendix A).

We inoculated two-week-old bread wheat seedlings with similar growth status overexpressed *TaRS15-3B* and wild-type seedlings with FES buffer (Mock), BSMV: γ-*TaRS15*, BSMV: γ-*PDS*, and BSMV: γ. We repeated the infection once a week later. Bleaching of the leaves of BSMV: γ-*PDS* was observed periodically, with significant photobleaching on day 14, while the leaves of plants inoculated with BSMV: γ as well as BSMV: γ-*TaRS15* showed slight streaks and symptoms of greenish discoloration (Figure 22a,b). The results of RT-qPCR on their leaves also showed that the expression levels of TaRS15 in the leaves of bread wheat seedlings inoculated with BSMV: γ-*TaRS15* were significantly lower than those of Mock- and BSMV: γ-inoculated bread wheat seedlings (Figure 22c).

#### 2.3.2. Identification of Drought Resistance Function of TaRS15-3B Gene-Silenced Plants

At this time, the bread wheat seedlings were treated with drought and salt stress. After 12 days of drought (Figure 23), they all showed symptoms of dehydration such as wilting and chlorosis of the leaves. The leaves of wild-type bread wheat seedlings showed obvious wilting, yellowing, and fading to a severe degree; however, the overexpression strain inoculated with BSMV: γ-*TaRS15* grew better than the recipient strain inoculated with BSMV: γ-*TaRS15*, and its performance did not differ significantly from that of bread wheat inoculated with Mock and BSMV: γ, which might be related to the BSMV-mediated silencing efficiency of the *TaRS15-3B* gene. After 3 days of rehydration, bread wheat overexpressing the *TaRS15-3B* gene showed a higher recovery capacity compared with Mock and BSMV: γ, while wild-type bread wheat seedlings inoculated with BSMV: γ-*TaRS15* showed a weaker recovery capacity. Furthermore, leaves of bread wheat seedlings inoculated with BSMV: γ-*TaRS15* showed a higher recovery capacity than those inoculated with BSMV: γ-*TaRS15* for rehydration after drought. In addition, bread wheat seedlings inoculated with BSMV: γ-*TaRS15* had a higher ability to recover from drought rehydration than wild-type bread wheat inoculated with BSMV: γ-*TaRS15*. This was probably due to the reduced efficiency of gene silencing after prolonged inoculation.

Before the drought treatment, there were no significant differences in the physiological indicators of bread wheat seedlings (Figure 24). After drought stress, compared with bread wheat inoculated with Mock and BSMV: γ, there was a highly significant decrease in MDA content and a highly significant increase in proline content, POD activity, CAT activity, and SOD activity in the *TaRS15-3B* overexpression strain, whereas for wild-type bread wheat seedlings inoculated with BSMV: γ-*TaRS15*, the results were the opposite of the overexpression strain, with MDA content showing a highly significant increase. In contrast, for wild-type bread wheat seedlings inoculated with BSMV: γ-*TaRS15*, the results were completely the opposite to those of the overexpression strain, with a highly significant increase in MDA content and a highly significant decrease in proline content, POD activity, CAT activity, and SOD activity. In addition, the overexpression strain inoculated with BSMV: γ-*TaRS15* showed a significant increase in SOD activity, but exhibited a significant decrease in POD activity and no significant changes in other physiological and biochemical parameters.

#### 2.3.3. Identification of Salt Resistance Function of TaRS15-3B Gene-Silenced Plants

After 18 days of salt stress (Figure 25), bread wheat seedlings showed obvious leaf curling, yellowing, the drying of leaf tips and growth inhibition. Compared with bread wheat inoculated with Mock and BSMV: γ, the overexpression of *TaRS15-3B* strain showed better growth status, with only slight leaf curling; bread wheat seedlings inoculated with BSMV: γ-*TaRS15* showed obvious leaf curling and the leaves of bread wheat seedlings inoculated with BSMV: γ-*TaRS15* showed obvious curling and yellowing, and the degree of wilting was severe. The BSMV: γ-*TaRS15* overexpression strain grew better than the wild-type bread wheat inoculated with BSMV: γ-*TaRS15* and showed little difference from the Mock- and BSMV: γ-inoculated bread wheat, which may be related to the efficiency of BSMV-induced silencing of the *TaRS15-3B* gene.

In the same way, there were no significant differences in the physiological indicators of bread wheat seedlings before the salt stress treatment (Figure 26). After treatment in salt stress conditions, the MDA content of the *TaRS15-3B* overexpression strain showed a highly significant decrease, while the proline content, POD activity, CAT activity, and SOD activity all showed highly significant increases compared with the Mock- and BSMV: γ-inoculated bread wheat. Conversely, for the BSMV: γ-*TaRS15*-inoculated wild-type bread wheat seedlings, the results were completely different from those of the overexpression strain. In contrast, the wild-type bread wheat seedlings inoculated with BSMV: γ-*TaRS15* showed a highly significant increase in MDA content and a highly significant decrease in proline content, POD activity, CAT activity, and SOD activity. In addition, the overexpression strain inoculated with BSMV: γ-*TaRS15* did not show significant changes in the physiological and biochemical parameters, except for a significant increase in SOD activity. It is speculated that this result may be due to the unstable efficiency of BSMV-induced *TaRS15-3B* gene silencing and the occurrence of an error during the experiment.

#### 2.3.4. Expression Levels of Stress Resistance-Related Genes in TaRS15-3B-Silenced Bread Wheat Plants

To further explore how *TaRS15-3B* is involved in the response of bread wheat to drought and salt stress, the expression levels of stress-related genes and ROS scavenging genes in *TaRS15-3B* silenced bread wheat plants were analyzed (Figure 27), and the results showed that, under drought as well as salt stress, the stress-related genes (*TaP5CS*, *TaLEA7*, and *TaDREB1*) and genes related to the ROS scavenging system (*TaPOD*, *TaSOD1*, and *TaCAT3*) were differentially down-regulated in bread wheat plants compared with Mock- and BSMV: γ-inoculated bread wheat plants. This suggests that the silencing of *TaRS15-3B* leads to a reduction in the ability of bread wheat plants to cope with stresses and ROS scavenging systems, which further increases their sensitivity to drought and salt.

## 3. Discussion

The raffinose synthase gene family has a variety of biological functions in plants [48,49,50]. The *RS* gene plays an important role in plant growth and development [38,39], which is essential for maintaining plant seed vitality and longevity [34,36]. In addition, it also plays an important role in plant resistance to abiotic stress [35,73,74,75]. *RS* gene families have been identified in multiple species [46,56,57,58,59]. Therefore, this study screened and identified *RS* family members in the whole-bread wheat genome, and analyzed their physicochemical properties, evolutionary relationships, conserved domains, promoter cis-acting elements, and expression patterns. On this basis, the functions of candidate gene *TaRS15-3B* under drought and salt stress were analyzed and evaluated.

In this study, a total of 34 genes encoding raffinose synthase were screened from the bread wheat genome. Members of the RS gene family all share the conserved structural domain of Raffinose_Syn (pf05691), which can be divided into four subfamilies. The *RS* gene family of bread wheat had more collinear blocks with the monocotyledonous plant *Aegilops tauschii Coss.*, and the evolutionary relationship was closer, indicating that the RS protein was conserved in the process of species evolution and the *RS* gene family in bread wheat could have been derived from the polyploidization process of bread wheat.

The results of cis-acting element analysis, GO enrichment analysis, and interaction protein prediction of the bread wheat *RS* gene family showed that the bread wheat *RS* gene family directly or indirectly participated in sugar metabolism in plants through ABA and MeJA signal transduction pathways, thus further responding to the abiotic stress of plants. There are differences in protein motifs, cis-acting elements, chromosomal localization, and expression levels in different tissues and abiotic stresses of bread wheat among subfamilies of the *RS* gene family, indicating that each subfamily performs different functions.

*TaRS13-3A*, *TaRS15-3B*, and *TaRS17-3D* on chromosome 3 of the bread wheat RS III subfamily have specific expression patterns, and their protein similarities are as high as 98.38%. The expression of the *TaRS15-3B* gene was relatively high in leaves and seeds and was up-regulated under drought, salt, ABA, and MeJA treatments. Therefore, *TaRS15-3B* was selected as a candidate gene for the next study. The molecular formula of *TaRS15-3B* protein is C_3804_H_5863_N_1037_O_1110_S_35_; the theoretical isoelectric point PI is 5.36, which belongs to acidic protein. The aliphatic amino acid index was 84.51, showing strong thermal stability; the average hydrophobicity was −0.074, indicating that the protein belongs to a hydrophilic protein; and the instability coefficient was 33.92 < 40, indicating that it belongs to a stable protein. In addition, TaRS15-3B had multiple potential phosphorylation sites, indicating that the protein is likely to participate in a signaling pathway through phosphorylation modification. However, the TaRS15-3B protein does not have any transmembrane structure or signal peptide sites. It is presumed to function only in the cytoplasm or organelles, consistent with both the results of subcellular localization and the findings that acidic α-galactosidase is located in protein storage vesicles in pea seeds [76].

The germination potential and germination rate of overexpressing *TaRS15-3B* gene line were higher than those of the wild-type strain under drought and salt stress conditions. At the germination stage, the *TaRS15-3B* gene caused the transgenic bread wheat to show higher tolerance to drought and salt stress. At the seedling stage, the overexpressing *TaRS15-3B* gene line showed significantly higher drought and salt tolerance, lower leaf wilt, and greater recovery ability than the wild-type strain. Similarly, after drought and salt stress, the MDA content of the overexpressing *TaRS15-3B* gene line decreased significantly, while proline content, POD activity, CAT activity, and SOD activity increased significantly, indicating a higher resistance to drought and salt stress.

At the later stage of bread wheat plant growth and development, after encountering drought stress, compared with the overexpressing *TaRS15-3B* gene line, wild-type bread wheat showed more obvious dehydration symptoms, plant height and spikelet number decreased significantly, and flag leaf length and effective tiller number decreased significantly under drought stress. In addition, the *TaRS15-3B* gene expression level showed a gradual upward trend in the middle and late stages of bread wheat growth and development. In addition, compared with normal water conditions, drought stress led to the significant upregulation of *TaRS15-3B* expression in various periods, especially in the heading stage and filling stage. The heading and filling stage, as the key period of bread wheat ear development, is the decisive period that affects the yield and quality of bread wheat. In addition, previous studies showed that *TaRS15-3B* gene expression was the highest in bread wheat seeds. It is presumed that it may further lead to the increase in raffinose content in seeds, thus showing higher germination vitality under drought and salt stress. The above results showed that the overexpressing *TaRS15-3B* gene line bread wheat in the middle and late stages of growth and development caused obvious resistance to drought stress and may improve seed vigor by increasing the raffinose content in seeds. However, further verification of this inference is needed.

To further clarify the role of *TaRS15-3B* in bread wheat response to drought and salt stress, *TaRS15-3B* gene silencing, induced by barley stripe mosaic virus (BSMV), was used. After drought and salt stress, *TaRS15-3B* gene-silenced bread wheat seedlings showed dehydration symptoms such as a loss of resistance to drought and salt stress, as well as general leaf wilting and chlorosis. The physiological indexes (MDA content, PRO content, POD activity, CAT activity, and SOD activity) of bread wheat seedlings also verified their sensitivity to drought and salt stress. In addition, in *TaRS15-3B*-silenced bread wheat plants, stress resistance-related genes (*TaP5CS*, *TaLEA7*, *TaDREB1*) [77,78,79] and ROS scavenging system-related genes (*TaPOD*, *TaSOD1*, and *TaCAT3*) [80,81,82] were down-regulated to varying degrees. The results showed that the silencing of the *TaRS15-3B* gene would lead to the decline of the ability of bread wheat plants to cope with stress and ROS scavenging system, thus further showing the enhanced sensitivity to drought and salt.

In this study, only the function of selected *TaRS15-3B* was analyzed under drought and salt stress, and further investigation is needed to determine how it functions. For example, three homologs of *TaRS720-5A*, *TaRS23-5B*, and *TaRS25-5D* on chromosome 5 showed increased expression under low-temperature stress, which is consistent with the association of cottonseed sugar with a response to cold stress in Arabidopsis and rice [48,83]. Meanwhile, the expression patterns of most of the *RS* family members in different tissues, as well as under abiotic stresses, showed consistency with the chromosome set. However, this needs to be verified by more evidence.

## 4. Materials and Methods

### 4.1. Plant Materials, Growth Conditions, and Stress Treatments

#### 4.1.1. Plant Materials

The bread wheat varieties were the cultivars KN199 and Chinese Spring (CS) and the T2 generation seed of KN199 overexpressing *TaRS15-3B*, both of which are kept in our laboratory.

#### 4.1.2. Growth Conditions, and Stress Treatments

Chinese Spring bread wheat seeds were soaked in 1.5% H_2_O_2_ for 24 h to break dormancy. They were then placed on moist filter paper to germinate them and treated with natural drought, NaCl solution, ABA hormone, and MeJA hormone until the two-leaf stage. Samples were taken at 0, 1, 3, 6, 9, 12, 24, and 48 h after treatment for subsequent trials.

The gene expression levels of the *TaRS15-3B* gene were examined in the leaves of the recipient bread wheat material KN199 and bread wheat overexpressing the *TaRS15-3B* gene, and three T3 generation line with high expression levels were selected for drought and salt tolerance functional identification.

The method used to identify the drought and salt tolerance of bread wheat at the germination stage was as follows: 30 intact and uniformly sized seeds of transgenic bread wheat and the control were selected, sterilized, and placed in Petri dishes lined with filter paper, to which deionized water (control), 20% PEG solution, and 200 mM NaCl solution were added in the corresponding treatments. The germination was counted and recorded.

The drought and salt tolerance of bread wheat was identified as follows: seedlings of uniform growth and good condition were selected and transferred to hydroponic boxes for one week. This corresponded to the addition of deionized water (control), 20% PEG solution, and 200 mM NaCl solution to simulate drought and salt stress. The boxes were placed in a light incubator (diurnal temperature 22 °C/20 °C, 16 h of light/8 h of darkness).

Soil culture was developed in the following way: we selected seedlings of uniform growth and in good condition and transplanted them into nutrient soil. Stress treatment was applied after one week of growth in the incubator. To achieve drought stress, we stopped watering the bread wheat and rehydrated it after a week when the leaves wilted. Salt stress treatment was performed as follows: we watered the bread wheat seedlings with 200 mM NaCl solution, observed growth, and replenished the NaCl solution promptly.

The identification of drought resistance and salt tolerance in the middle and late stages of bread wheat growth and development was conducted as follows: The transgenic bread wheat seedlings and the control were selected and transplanted in plastic buckets at the start of the bread wheat growing period and then subjected to two water treatments. The first was called ‘normal water supply’ (WW): in this method, we provided an adequate water supply throughout the reproductive period, i.e., soil moisture content controlled to (75 ± 5)% of the maximum field water holding capacity. The second was termed ‘drought stress’ (WD): soil moisture content before the pulling period was the same as that of the normal water supply group, and after the pulling period we started the water control at (45 ± 5)% of the maximum field water holding capacity and maintained this level until maturity.

### 4.2. Identification of Gene Family Members

In this study, the genome sequence and genome annotation information of bread wheat were downloaded from the Ensemble Plants database [84], and a local bread wheat protein database was constructed using TBtools software v1.125 [85]. The sequences of *RS* gene family related proteins in Arabidopsis were then downloaded from the TAIR database [86], and two-way BLAST matching was performed using TBtools software v1.125 and the NCBI database [87] to obtain family members. In addition, the hidden Markov model (HMM) of PF05691 was downloaded from the Pfam database [88]. The preliminary screening of bread wheat proteins based on the HMM model of cottonseed glycan synthetase RS protein was performed using HMMER v3.3.2 software to obtain *TaRS* family members, combining the results of the two-way BLAST as well as the HMMER search. After removing redundancy, only their sequence results containing specific conserved structural domains were obtained. Finally, the screened structures were reconstructed into the HMM model using HMMER hmmbuild, and the bread wheat protein database was screened again to obtain all the members of the bread wheat gene family. Subsequently, the physicochemical properties of TaRS proteins were predicted on the ExPASY website [89]. The WOLF PSORT website was used to predict where they functioned in the cell [90].

### 4.3. Classification and Phylogenetic Analysis

In this study, the genome sequences and genome annotation information of *Aegilops tauschii Coss*, *Oryza sativa Japonica*, *Zea mays*, *Arabidopsis thaliana*, *Glycine.max*, *Gossypium raimondii*, *Solanum tuberosum* L., and *Solanum lycopersicum* L. were downloaded from the Ensembl Plants database [84], and a local protein database was constructed using TBtools software v1.125 [85]. HMMER search software was used to identify RS proteins in various species based on the self-built HMM model of RS protein of bread wheat raffinose synthase. A phylogenetic tree was constructed through the use of MEGA X software, and the evolutionary tree was visualized and modified through iTOL website [91].

### 4.4. Gene Structure and Conserved Motif Analysis

The TaRS protein was submitted to MEME (http://meme-suite.org, accessed on 25 December 2022) [92] to retrieve conserved motifs in it and the conserved structural domains of the TaRS protein were viewed through the NCBI Protein Batch CD-search database [93]. The gene structure was visualized using TBtools software v1.125 based on information from bread wheat genome annotation files and genome files [85].

### 4.5. Chromosomal Location and Identification of Homoeologs

Information on the location of the *TaRS* gene was obtained from the bread wheat genome annotation file and the distribution of the *TaRS* gene on the chromosome was mapped using TBtools software v1.125 [85]. The identification of homologous genes was completed using the Triticeae-GeneTribe database (http://breadwheat.cau.edu.cn/TGT/faq.html, accessed on 15 January 2023) [62].

### 4.6. Collinearity and Evolutionary Selection Analysis

All TaRS proteins were searched based on protein BLASTP, and duplicate gene pairs were defined according to the following criteria [61]: (1) the sequence length of comparable pairs of protein sequences exceeded 80% of the longer protein sequence; (2) regional similarity in the pair was >80%; and (3) closely linked genes were counted only once [94]. The TBtools software v1.125 was used to determine fragment duplication events and tandem repeat events in the bread wheat *TaRS* gene and to visualize duplicate gene pairs in the bread wheat genome [85].

The TBtools software v1.125 was used to determine the co-linear relationships between RS gene family members in bread wheat and other species and to calculate the rates of non-synonymous and synonymous substitutions and their respective ratios for each pair of homologous gene pairs in bread wheat with *Aegilops tauschii Coss*, *Oryza sativa Japonica*, *Glycine.max*, and bread wheat itself [85]. Violin plots of Ka/Ks ratios were plotted in GraphPad Prism 8.

### 4.7. Cis-Acting Element Analysis

The 2000 bp sequence upstream of the start codon of the *TaRS* gene was used as the promoter region [95]. The sequences in the promoter region were identified by the PlantCARE database (http://bioinformatics.psb.ugent.be/webtools/plantcare, accessed on 3 February 2023) as having cis-acting elements to infer the possible biological functions and transcriptional regulation of the *TaRS* gene [96].

### 4.8. Gene Ontology (GO) Analysis and Protein Interaction Network (PPI) Analysis

GO enrichment analysis of *TaRS* gene family members was performed through the AGriGO database (http://systemsbiology.cau.edu.cn/agriGOv2/FAQ.php, accessed on 3 February 2023) [97]. In addition, the amino acid sequences of *TARS* and its corresponding Arabidopsis homologs were submitted to the STRING database (http://string-db.org/cgi, accessed on 10 March 2023) [98] for prediction of its protein–protein interaction networks (PPI networks), and then visualized by the Cytoscape v3.9.1. software [99].

### 4.9. Characterization of Expression under Different Tissue and Abiotic Stresses

In order to investigate the expression of *TaRS* gene members in different tissues of bread wheat at different developmental periods and under abiotic stress, we downloaded gene expression data for members of this family from the bread wheat expression database (http://www.breadwheat-expression.com, accessed on 12 March 2023) [100] and plotted expression heat maps.

### 4.10. RNA Extraction and Quantitative Real-Time PCR

Total RNA was isolated from bread wheat leaves using TRIGene (GenStar, Beijing, China). The EasyScript^®^One-Step gDNA Removal and cDNA Synthesis SuperMix (TransGen Biotech, Beijing, China) was used for cDNA synthesis. QuantStudio 3 Flex Real-Time PCR system (ThermoFisher, Foster City, CA, USA) and PerfectStart® Green qPCR SuperMix (+Universal Passive Reference Dye) (TransGen Biotech, Beijing, China) were used for quantitative real-time PCR (qRT-PCR). The bread wheat β-actin gene (GenBankaccession number AB181991.1) was used as an internal reference for all qRT-PCR analyses. Three repetitions were set for each sample. The relative expression levels of each gene were calculated based on the 2^−△△CT^ value. The primers used in this experiment are shown in Appendix A.

### 4.11. Determination of Physiological and Biochemical Indicators

The main components were malondialdehyde (MDA), proline (Pro), peroxidase (POD), catalase (CAT), and superoxide dismutase (SOD). The assays were performed using physiological biochemical kits from Beijing Solebro Technology Co., Ltd. (Beijing, China) according to their instructions, with three biological replicates used for each physiological indicator.

### 4.12. Generation and Identification of Overexpression Lines in Wheat

The transformation of wheat healing tissue was performed using the gene gun binary vector transformation method. The overexpression vector for wheat was pWMB003, the glufosinate (PPT) resistance screening vector was pAHC20 and the wheat material was KN199. To obtain T0 generation wheat overexpression lines:(1)Young ears of wheat that had been pollinated for about 10 days were taken back and sterilized; the young embryos were picked out, placed in an induction medium and incubated away from light to induce guilted tissues.(2)We transferred the healing tissues from the previous step to a hypertonic medium for 6 h. Afterwards, we bombarded the healing tissues with the gene gun method and continued to incubate them in the hypertonic medium for 16 h. Then, we changed to an induction medium and continued to incubate the tissues away from light.(3)The healing tissues obtained in step 2 were replaced in a differentiation medium (with PPT), after which the differentiated seedlings were transferred to a strong medium for further cultivation and vernalization, and were finally transplanted to the soil for further cultivation.(4)After obtaining the T3 overexpression lines, those with higher expression were identified by PCR and qPCR for the next step of functional identification of the transgenic wheat.

### 4.13. BSMV-Induced Gene-Silencing of TaRS15-3B

The recombinant viral vectors BSMV: γ-*TaRS15* and BSMV: γ-*PDS* and the viral vectors BSMV:α, BSMV:β, and BSMV: γ were linearized after linking the *TaRS15* gene-specific fragment to the BSMV: γ vector. The above purified and recovered linearized recombinant viral vectors were transcribed in vitro using the T7 RiboMAX™ Express Large Scale RNA Production System (Promega, Madison, WI, USA). The in vitro transcript products were mixed 1:1:1 and then added to the FES buffer at 3:22 and mixed well to form the inoculum. When the bread wheat seedlings reached the three-leafed stage, 20 µL of inoculum was inoculated on the third leaf of the bread wheat seedlings. Bread wheat seedlings that had been inoculated with 1xFES buffer were used as a control (Mock). After inoculation, the bread wheat was placed at room temperature and protected from light for 24 h and kept moist. The bread wheat seedlings were then grown in a light incubator at 23−25 °C with a photoperiod of 16 h/8 h (light/dark) and the infestation was repeated after one week. Some 10 to 14 days after virus inoculation, thread-through photobleaching can be observed on bread wheat leaves inoculated with BSMV: γ-*PDS*. The third leaf of bread wheat seedlings inoculated with the target gene was taken and RT-qPCR was performed to detect the expression of *TaRS15* to determine the silencing efficiency of the target gene. Mock and the infested bread wheat were then subjected to drought and salt stress treatments.

### 4.14. Statistical Analysis

Statistical analysis of the data was performed using Microsoft Excel 2019 software and ANOVA was performed using IBM SPSS Statistics 25 statistical analysis software. The significance levels were defined as * (*p* < 0.05), ** (*p* < 0.01), and *** (*p* < 0.001). The data were plotted using GraphPad Prism 8.0.

## 5. Conclusions

Based on the bread wheat genome information, a total of 34 genes encoding raffinose synthase were screened, and their physicochemical properties, evolutionary relationships, conserved domains, promoter cis-acting elements, and expression patterns were systematically analyzed. On this basis, we found that *TaRS15-3B* can improve the drought and salt tolerance of bread wheat, providing direction for the study of other bread wheat *RS* gene families and candidate genes for the genetic improvement of bread wheat.

## Figures and Tables

**Figure 1 ijms-24-11185-f001:**
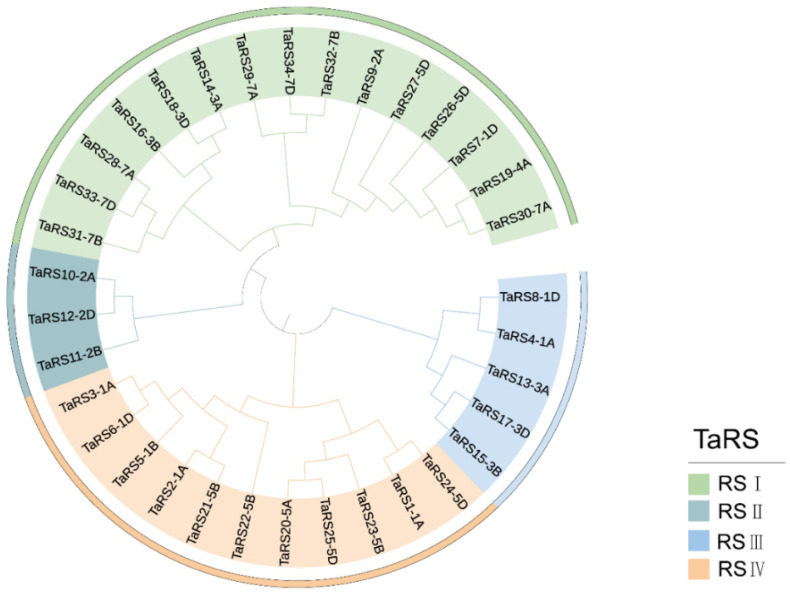
Phylogenetic tree of TaRS protein in *Triticum aestivum*. These subfamilies are represented by different colors: RS I (green), RS II (cyan), RS III (blue), and RS IV (orange yellow).

**Figure 2 ijms-24-11185-f002:**
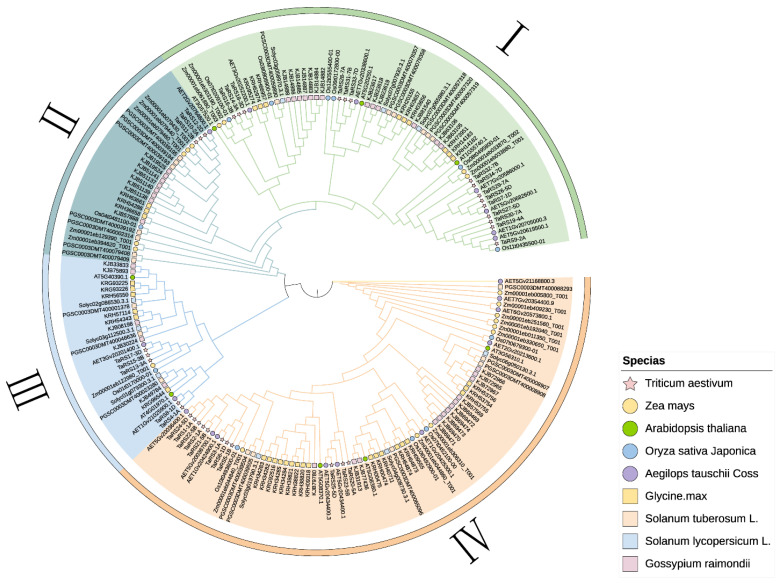
A phylogenetic tree of RS protein in monocotyledons (*Triticum aestivum*, *Aegilops tauschii Coss*, *Oryza sativa Japonica*, and *Zea mays*) and dicotyledons (*Arabidopsis thaliana*, *Glycine.max*, *Gossypium raimondii*, *Solanum tuberosum* L. and *Solanum lycopersicum* L.). These subfamilies are represented by different colors: RS I (green), RS II (cyan), RS III (blue), and RS IV (orange yellow). The monocotyledons and dicotyledons are represented by circles and squares, respectively. The differently colored shapes represent different species.

**Figure 3 ijms-24-11185-f003:**
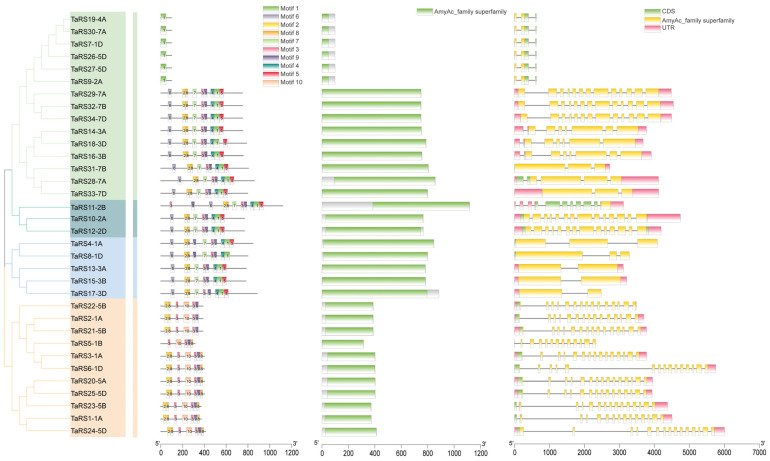
Phylogenetic analysis of TaRS proteins in bread wheat, conserved motifs, conserved domain, and gene structure.

**Figure 4 ijms-24-11185-f004:**
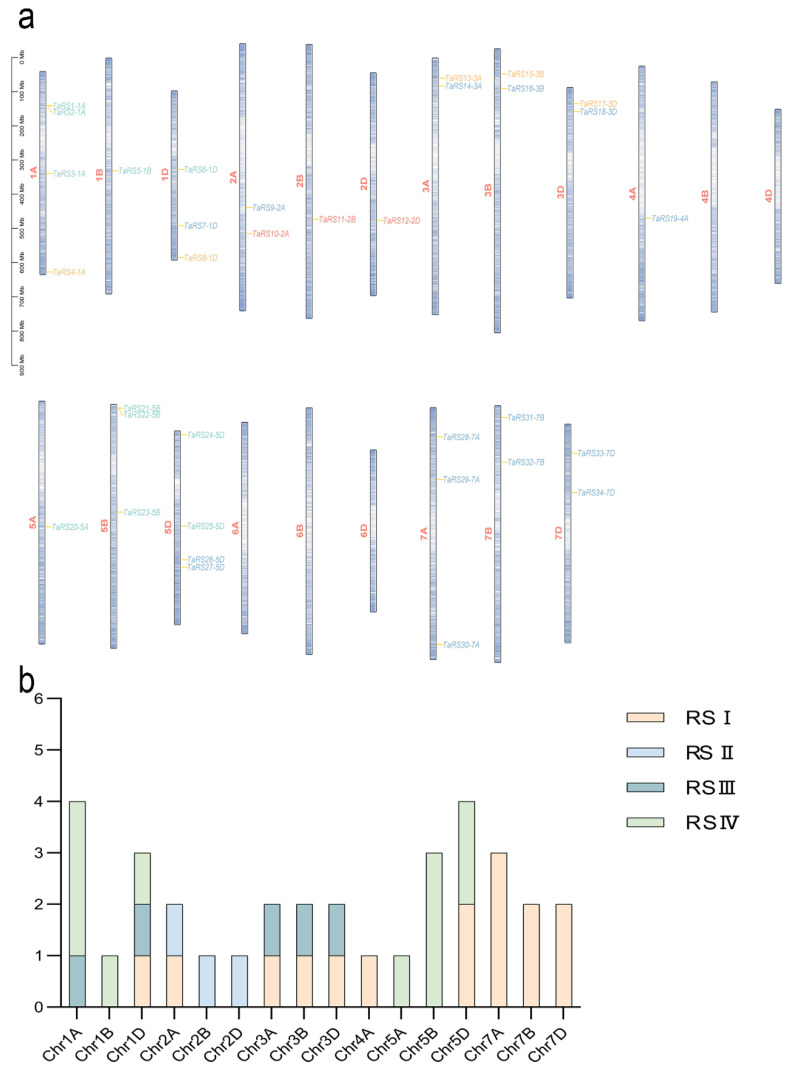
The distribution of *TaRS* genes on bread wheat chromosomes. (**a**) The number of *TaRS* genes on different chromosomes (Chr1–Chr7). These subfamilies are represented by different colors: RS I (green), RS II (cyan), RS III (blue), and RS IV (orange yellow). (**b**) The number of *TaRS* genes per chromosome: the darker the color of the chromosome, the higher the gene density at that location. These subfamilies are represented by different colors: RS I (green), RS II (cyan), RS III (blue), and RS IV (orange yellow).

**Figure 5 ijms-24-11185-f005:**
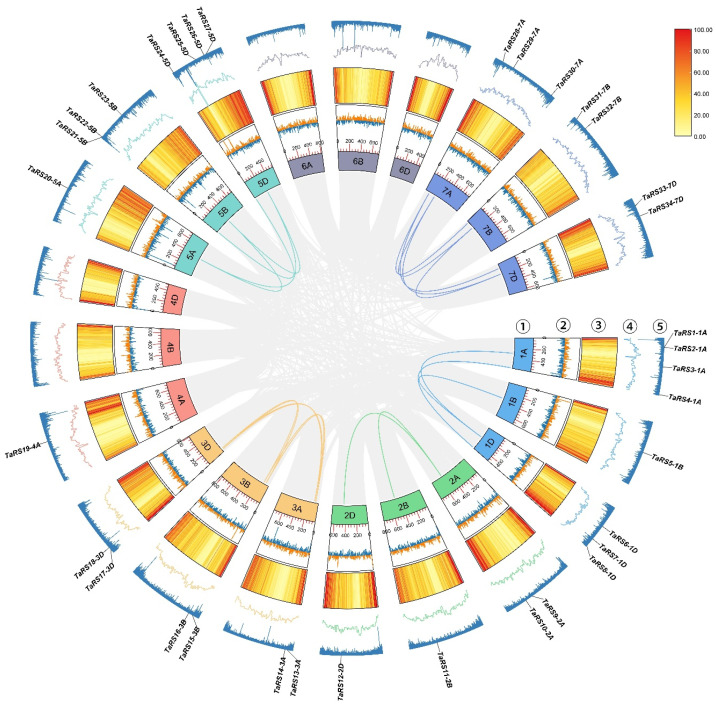
Collinearity analysis of the *TaRS* gene family in the bread wheat genome. The segment and tandem repeats of the *TaRS* genes are mapped out in the bread wheat genome. Grey lines represent all duplicate gene pairs in the bread wheat genome, while colored lines represent duplicate gene pairs between *TaRS* genes on different chromosomal groups. ①: bread wheat chromosome group, whereby different chromosome groups are represented by different colors; ②: GCskew; ③: bread wheat gene density. Different colors represent different expression values: red: higher expression; yellow: lower expression; ④: GCratio; ⑤: Nratio.

**Figure 6 ijms-24-11185-f006:**
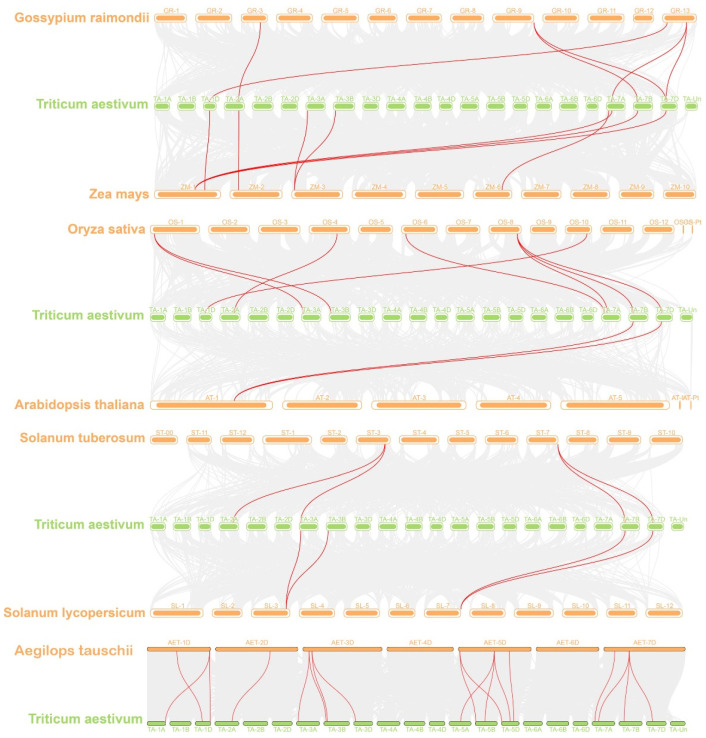
Syntenic relationships of the *TaRS* genes in bread wheat and other species. Genomic collinearity regions of bread wheat and other species are indicated by gray lines. The red lines indicate the syntenic *TaRS* gene pairs.

**Figure 7 ijms-24-11185-f007:**
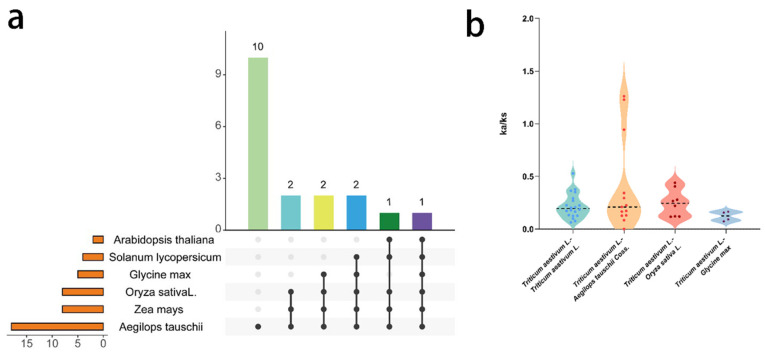
Syntenic and evolutionary analyses in bread wheat TaRS family. (**a**) UpSet plot of non-redundant TaRS genes in different species. (**b**) Violin plot of Ka/Ks rations in duplicated *TaRS* gene pairs.

**Figure 8 ijms-24-11185-f008:**
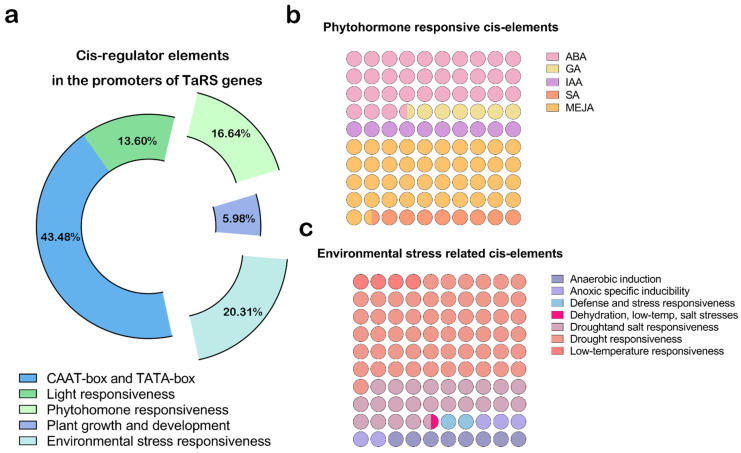
The analysis results of *TaRS* genes promoter sequence (2000 bp upstream of genes). (**a**) The percentage distribution of cis-regulator elements in the promoters of *TaRS* genes. (**b**) The percentage distribution of phytohormone-responsive cis-elements. (**c**) The percentage distribution of environmental stress-related cis-elements. The area of the circle represents the number of cis-acting elements.

**Figure 9 ijms-24-11185-f009:**
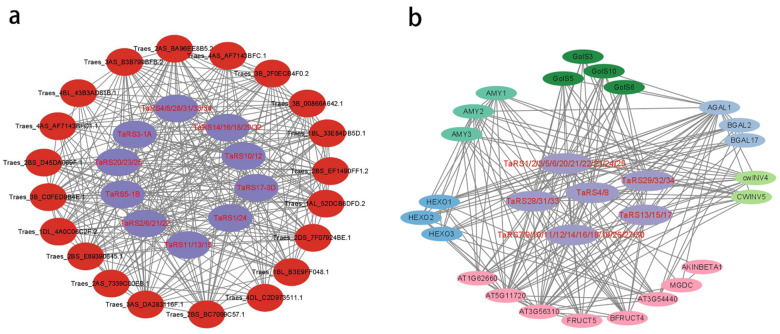
Predicted protein–protein interaction networks of *TaRS* genes with other bread wheat proteins using the STRING database. (**a**) TaRS protein interaction network in bread wheat; (**b**) TaRS homologous protein interaction network in *Arabidopsis thaliana*. The purple ovals represent TaRS homologous proteins in Arabidopsis, while the grey lines represent protein interactions.

**Figure 10 ijms-24-11185-f010:**
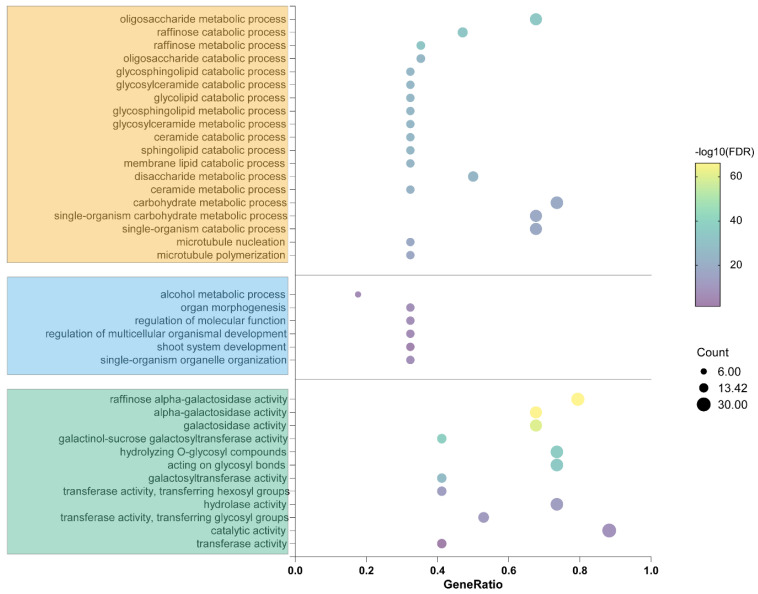
*TaRS* gene ontology analysis. Orange, blue, and green represent molecular function, cellular composition, and biological processes, respectively.

**Figure 11 ijms-24-11185-f011:**
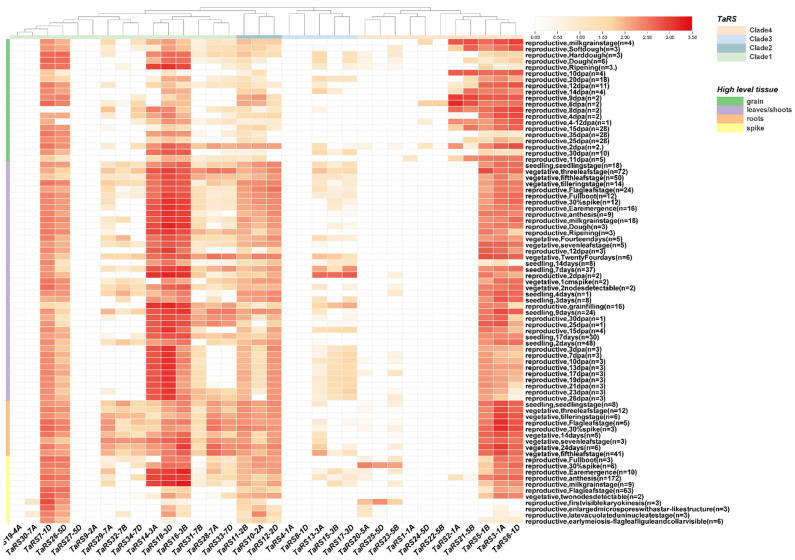
The expression of *TaRS* genes during bread wheat developmental stages of different tissues (based on Log_2_(tpm + 1)). The heatmap shows the *TaRS* genes expression levels: abscissa represents different genes and ordinate represents different growth periods of different tissues. Different colors represent different expression values, as red: higher expression; white: lower expression.

**Figure 12 ijms-24-11185-f012:**
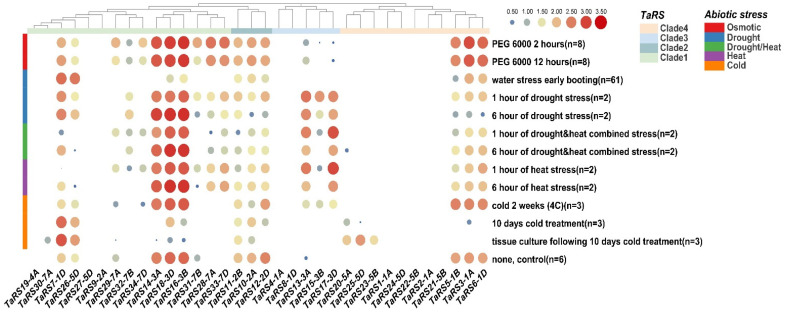
The expression of *TaRS* genes in bread wheat under abiotic stress (based on Log_2_(tpm + 1)). Different colors and sizes represent different expression values: blue to red, and the graph area from small to large represents the expression level from low to high.

**Figure 13 ijms-24-11185-f013:**
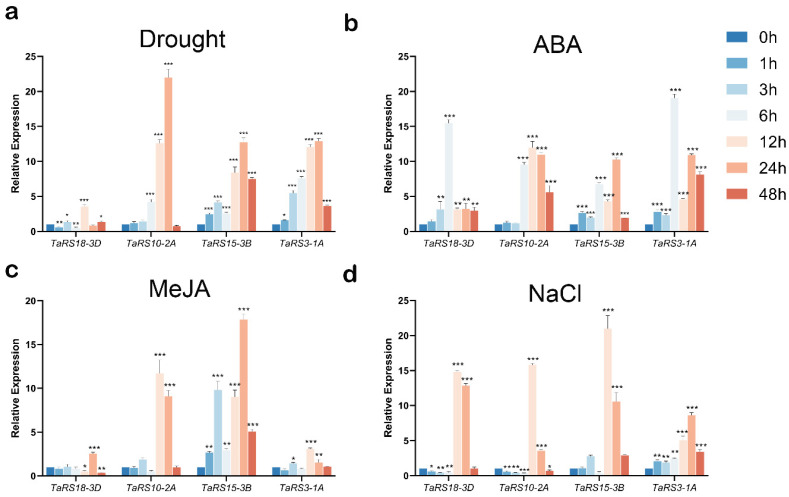
The real-time quantitative PCR analyses of four *TaRS* genes under (**a**) drought, (**b**) ABA, (**c**) MeJA, and (**d**) NaCl treatments. The significance levels were defined as * (*p* < 0.05), ** (*p* < 0.01), and *** (*p* < 0.001).

**Figure 14 ijms-24-11185-f014:**
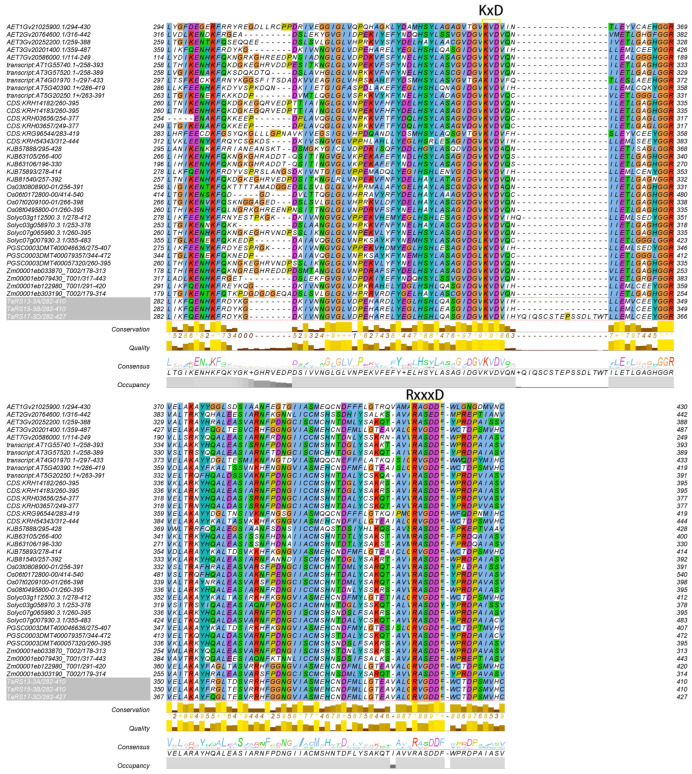
Multiple sequence alignment of the conserved RS domain.

**Figure 15 ijms-24-11185-f015:**
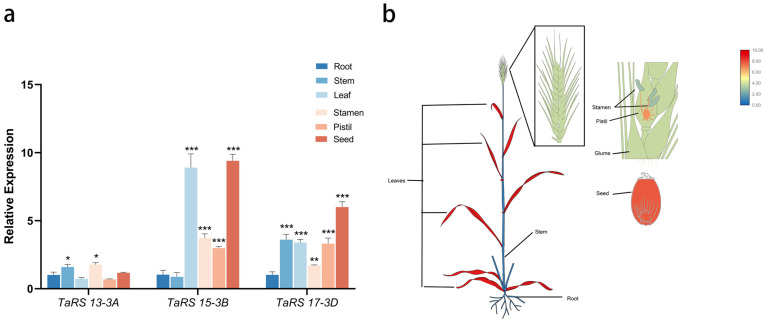
Expression levels in different tissues of bread wheat: (**a**) Relative expression levels of TaRS13-3A, TaRS15-3B, and TaRS17-3D in different tissues of bread wheat; (**b**) heat map of TaRS15-3B expression in different tissues, different colors represent different expression values: red: higher expression levels; blue: lower expression levels. The significance levels were defined as * (*p* < 0.05), ** (*p* < 0.01), and *** (*p* < 0.001).

**Figure 16 ijms-24-11185-f016:**
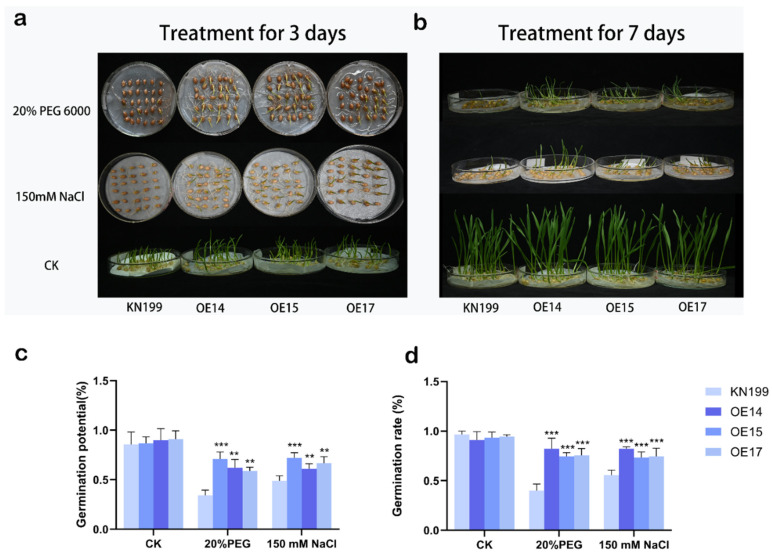
Identification of bread wheat germination resistance: (**a**) performance after three days of stress treatment; (**b**) performance after seven days of stress treatment; (**c**) germination potential statistics; (**d**) germination rate statistics. The significance levels were defined as ** (*p* < 0.01), and *** (*p* < 0.001).

**Figure 17 ijms-24-11185-f017:**
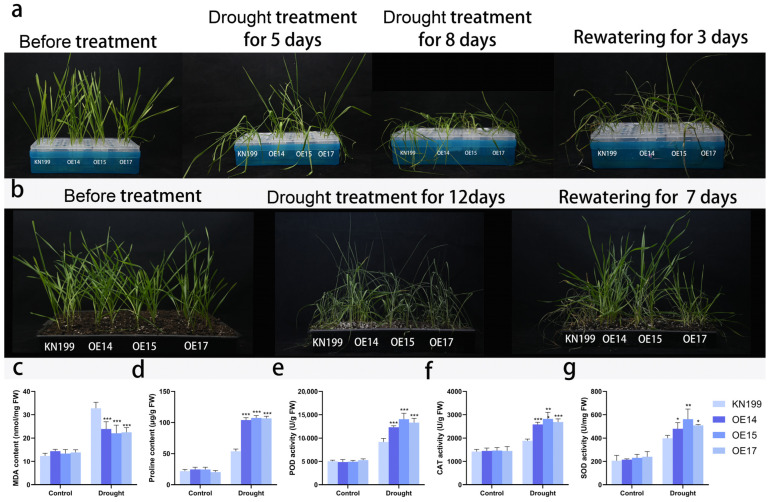
Identification of drought resistance in bread wheat seedlings: (**a**) phenotype before and after stress by hydroponics; (**b**) phenotype before and after stress by soil culture; (**c**) MDA content; (**d**) proline content; (**e**) POD activity; (**f**) CAT activity; (**g**) SOD activity. The significance levels were defined as * (*p* < 0.05), ** (*p* < 0.01), and *** (*p* < 0.001).

**Figure 18 ijms-24-11185-f018:**
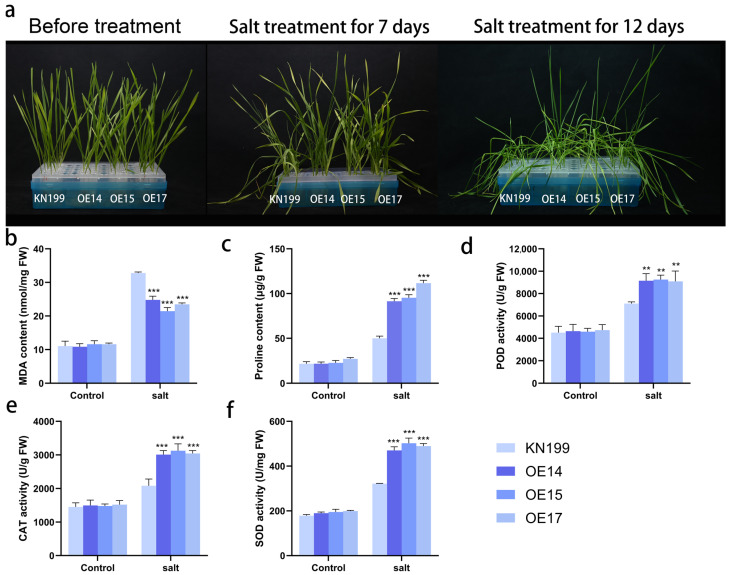
Identification of salt resistance in bread wheat seedlings: (**a**) phenotype before and after stress by hydroponics; (**b**) MDA content; (**c**) proline content; (**d**) POD activity; (**e**) CAT activity; (**f**) SOD activity. The significance levels were defined as ** (*p* < 0.01), and *** (*p* < 0.001).

**Figure 19 ijms-24-11185-f019:**
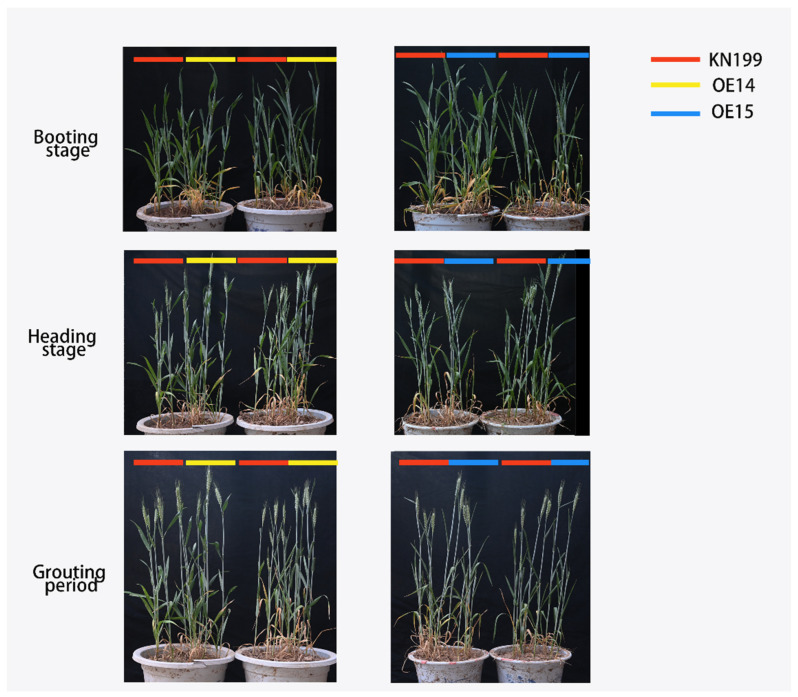
Drought assay of TaRS15-3B overexpression lines in different development stage of wheat: booting-stage phenotype; heading-stage phenotype; grouting phenotype.

**Figure 20 ijms-24-11185-f020:**
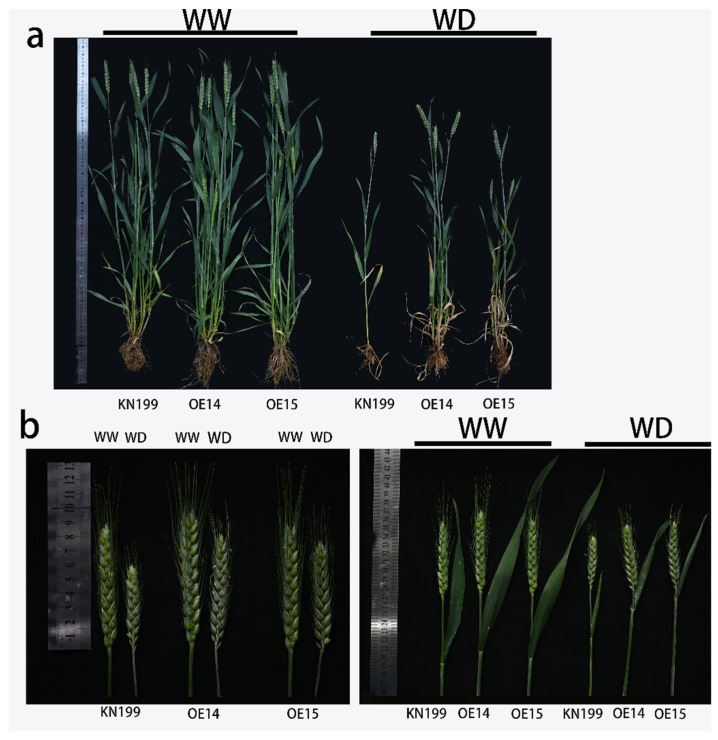
Identification of stress resistance function in bread wheat in the middle and late stages of development: (**a**) plant phenotype in the grouting phenotype; (**b**) ear phenotype during grouting phenotype.

**Figure 21 ijms-24-11185-f021:**
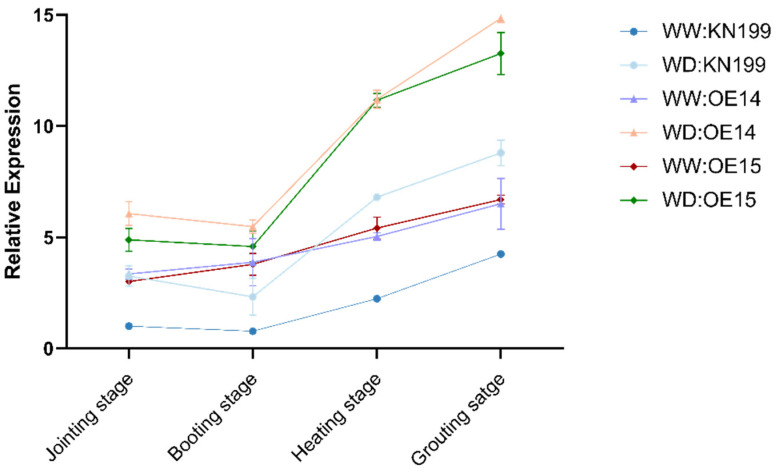
Expression levels of the *TaRS15-3B* gene in bread wheat at the jointing stage, booting stage, heading stage, and grouting stage.

**Figure 22 ijms-24-11185-f022:**
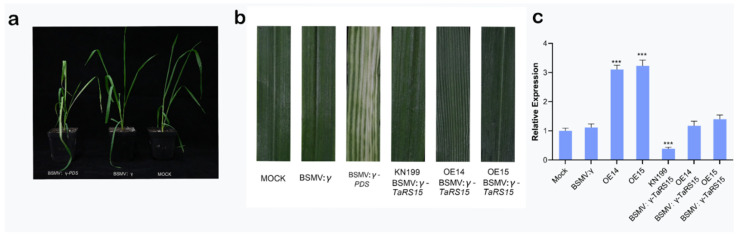
BSMV-mediated silencing of the TaRS15-3B gene: (**a**) plant phenotype after BSMV virus inoculation; (**b**) leaf phenotype after BSMV virus inoculation: (**c**) silencing efficiency of the TaRS15-3B gene. The significance levels were defined as *** (*p* < 0.001).

**Figure 23 ijms-24-11185-f023:**
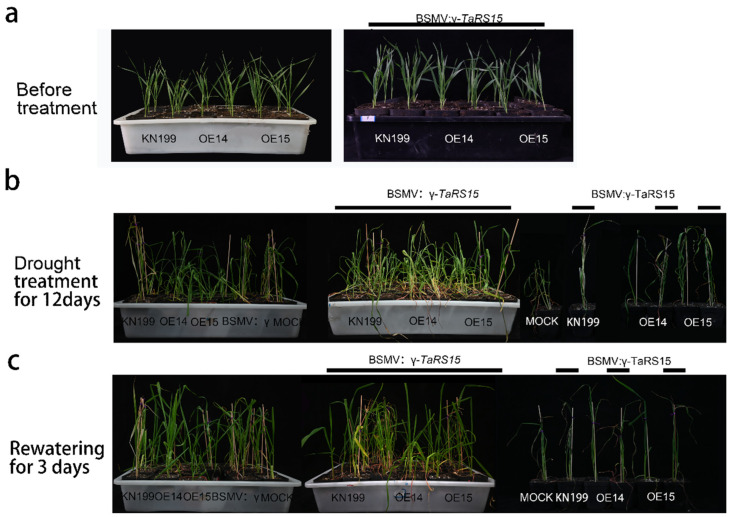
Identification of drought resistance in TaRS15-3B gene-silenced plants: (**a**) phenotype before drought stress; (**b**) phenotype 12 days after drought stress (**c**); phenotype analysis 3 days after rehydration.

**Figure 24 ijms-24-11185-f024:**
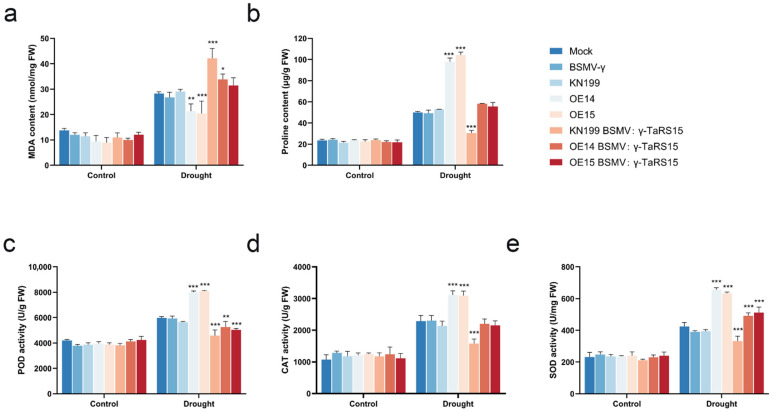
Physiological and biochemical indicators of silenced plants with TaRS15-3B gene: (**a**) MDA content; (**b**) PRO content; (**c**) POD activity; (**d**) CAT activity; (**e**) SOD activity. The significance levels were defined as * (*p* < 0.05), ** (*p* < 0.01), and *** (*p* < 0.001).

**Figure 25 ijms-24-11185-f025:**
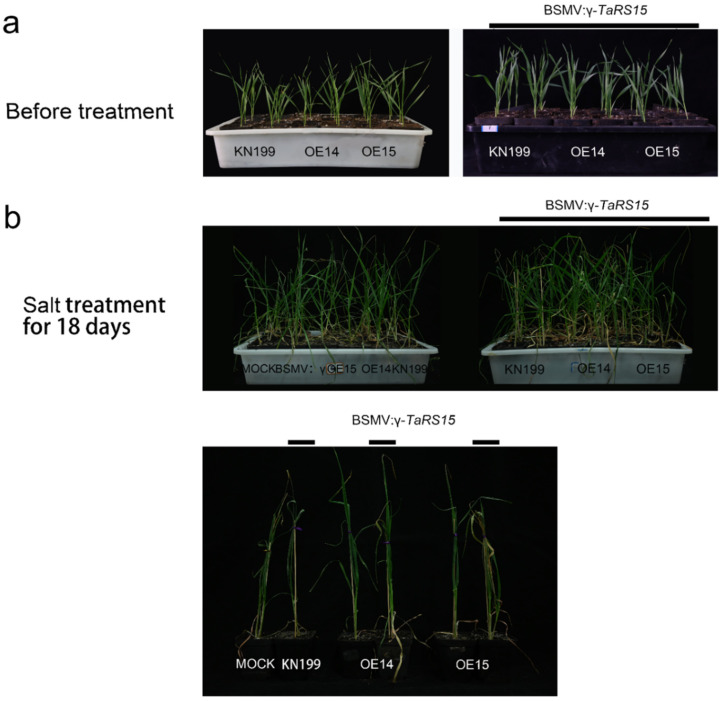
Identification of salt resistance in silenced plants with *TaRS15-3B* gene: (**a**) phenotypic analysis before salt stress; (**b**) phenotypic analysis before salt stress.

**Figure 26 ijms-24-11185-f026:**
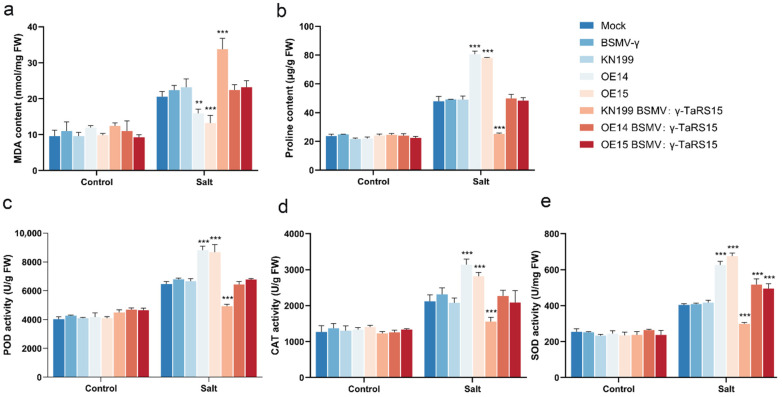
Identification of salt resistance in TaRS15-3B gene-silenced plants:(**a**) MDA content; (**b**) PRO content; (**c**) POD activity; (**d**) CAT activity; (**e**) SOD activity. The significance levels were defined as ** (*p* < 0.01), and *** (*p* < 0.001).

**Figure 27 ijms-24-11185-f027:**
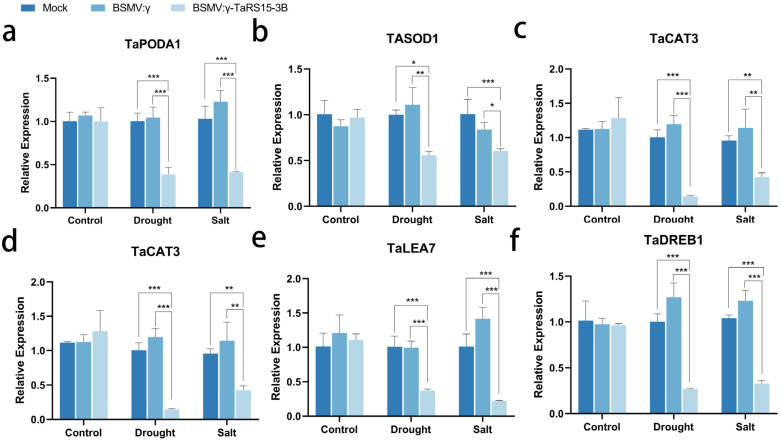
The relative expression levels of stress-related genes and ROS scavenging genes in the *TaRS15-3B* silencing bread wheat plants: (**a**) *TaPODA1*; (**b**) *TaSOD1*; (**c**) *TaCAT3;* (**d**) *TaP5CS*; (**e**) *TaLEA7*; (**f**) *TaDREB1*. The significance levels were defined as * (*p* < 0.05), ** (*p* < 0.01), and *** (*p* < 0.001).

**Table 1 ijms-24-11185-t001:** Homoeologous *TaRS* genes in bread wheat.

Homoeologous Group (A:B:D)	All Bread Wheat Genes	All TaRS Genes		
		Number of Groups	Number of Genes	% of Genes
1:1:1	35.80%	6	18	52.94%
1:1:n/1:n:1/n:1:1, n > 1	5.70%	1	4	11.76%
1:1:0/1:0:1/0:1:1	13.20%	3	6	17.65%
Orphans/singletons	37.10%	-	1	2.94%
Other rations	8.00%	2	5	14.71%
Total	99.80%	12	34	100%

**Table 2 ijms-24-11185-t002:** Investigation of agronomic traits at the late filling stage. The difference is not significant if there is a letter with the same marker, and significant if there is a letter with different markers. Generally, lower-case letters indicate a significant level of α = 0.05; upper case letters indicate a significant level of α = 0.01.

	Lines	Flag Leaf Length (cm)	Flag Leaf Width (cm)	Plant Height (cm)	Spikelets Number	Valid Tillering Number
WW	KN199	21.50 ± 0.76 a A	1.746 ± 0.08 a A	76.46 ± 1.01 a A	15.66 ± 0.57 ab AB	6.166 ± 1.16 a A
	OE14	21.69 ± 0.87 a A	1.793 ± 0.12 a A	76.00 ± 0.38 a A	17 ± 1 a A	6.333 ± 1.50 a A
	OE15	21.94 ± 0.39 a A	1.753 ± 0.10 a A	75.9 ± 0.46 a A	16.66 ± 1.52 ab A	5.833 ± 1.32 a A
WD	KN199	12.01 ± 0.26 cB	1.208 ± 0.09 b B	53.73 ± 1.34 c C	13.33 ± 0.57 c C	1.5 ± 0.83 c B
	OE14	13.14 ± 0.39 b B	1.385 ± 0.04 b B	60.23 ± 1.12 b B	16.33 ± 0.57 ab A	3.333 ± 0.81 b B
	OE15	12.88 ± 0.33 b B	1.385 ± 0.10 b B	60.46 ± 1.09 b B	14.66 ± 0.57 bc AB	2.666 ± 1.03 bc B

## Data Availability

Not applicable.

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
