# Peer review of "Analysis of Raffinose Synthase Gene Family in Bread Wheat and Identification of Drought Resistance and Salt Tolerance Function of *TaRS15-3B"

_ijms, 2023, doi:10.3390/ijms241311185_

Round 1
Reviewer 1 Report
The authors successfully conducted a comprehensive analysis of the raffinose synthase gene family in wheat, leading to the identification and characterization of the 3TaRS15-3B gene. They employed a combination of bioinformatics, molecular biology techniques, and functional assays to elucidate the role of this gene in drought resistance and salt tolerance. The study's findings shed light on the molecular mechanisms underlying wheat's response to abiotic stresses and contribute to the broader understanding of plant stress tolerance. The authors performed good analysis, writing, and presentation of the work. However, there is still a need for attention to improve the quality of the MS.
Specific comments:
To address the issue of spacing between words, some adjustments were suggested below, and the authors conducted a thorough review of the entire manuscript.
Line 14-15: rephrase the sentence for the intended meaning.
Line 741: system-related genes(TaPOD, TaSOD1, TaCAT3)in wheat
Line 745: BSMV:γ inoculated wheat plants. Need space between BSMV:γ
TBtools software[85]
raffinose synthase[88].
(https://itol.embl.de/)[91].
tation files and genome files[85].
served motifs in it[92].
were counted only once[94].
CARE database(http://bioinformatics.psb.ugent.be/webtools/plantcare)
Cytoscape v3.9.1. software[99].
(http://www.wheat-expression.com)[100].
(based on Log2(tpm+1)).
corn seed germination[43],
resistance of cereal seeds[41, 42]
was significantly reduced[40].
life[37].
There is Minor editing of English language required
Author Response
Dear Editors and Reviewers:
We feel great thanks for your professional review work on our article. As you are concerned, several problems need to be addressed. According to your excellent suggestions, we have made extensive corrections to our previous draft, the detailed modifications are listed below.
We feel sorry for our carelessness. In our resubmitted manuscript, the issue of spacing between words has been resolved. In addition, we do invite a friend of ours who is a native English speaker from the USA to help polish our article. And we hope the revised manuscript could be acceptable to you.
For Lines 14-15 in the article, We have rewritten this part according to the Reviewer’s suggestion. Finally, we have rechecked the relevant references you pointed out and made corresponding corrections. We were sorry for our careless mistakes. Thank you for your reminder.
We tried our best to improve the manuscript and made some changes marked in red in the revised paper which will not influence the content and framework of the paper. We appreciate Editors/Reviewers’ warm work earnestly and hope the correction will meet with approval. Once again, thank you very much for your comments and suggestions.

Reviewer 2 Report
In the manuscript Guo et al. carried out an interesting and complete characterization of the raffinose synthase gene family in bread wheat, including the study of expression patterns of TaRS genes in abiotic stress conditions followed by a functional characterization of the TaRS15-3B gene, as well as the identification of drought and salt tolerance function.
However, the manuscript should be improved as follows:
1. When the authors refer to wheat in the manuscript, they are talking about bread wheat. Therefore, it should be written as bread wheat throughout the whole article, in order to avoid misunderstanding with other wheat species such as durum wheat.
2. Figures 2, 3, 4A and 11 should be presented in a better resolution.
3. In Figure 15a the statistical significance of the expression levels should be added.
4. The scientific names of species and the names of genes should be written in italics, e.g. lines 104, 367, 241.
5. The whole manuscript should be re-read carefully by the authors, since some words are misspelt, e.g. line 276 “yellow” instead of “yollow”. Additionally, the meaning of some sentences is quite obscure, e.g. lines 439, 460, 496.
6. An odd symbol is used in many parts of the manuscript instead of the comma, e.g. lines 184, 185, 186, 198, 199, 200, 324.
7. In some cases, spaces should be added after comma, after points or between words, e.g. in lines 145, 146, 164, 186, 202, 290, 320, 365, 422, 669. On the other hand, there are some spaces that need to be deleted e.g. line 308.
8. Paragraph from line 520 to line 525 is not understandable. Authors should write again this part.
9. In line 394 authors referred to nutritional growth stage. Do they mean vegetative growth stage? This term should be changed.

1. The whole manuscript should be proof-read by a native English speaker, as many parts of the text could be improved, e.g. line 30 “livestock feed” instead of “animal feed species”, line 38 “. This may account” instead of “and is an important reason”, line 366 “the most” instead of “the more”.
Author Response
Dear Editors and Reviewers:
We feel great thanks for your professional review work on our article. As you are concerned, several problems need to be addressed. According to your excellent suggestions, we have made extensive corrections to our previous draft, the detailed modifications are listed below.
1. We sincerely thank the reviewer for careful reading. As suggested by the reviewer, we have corrected the “ wheat ” into “ durum wheat ”.
2. We have reset the Figures to ensure better resolution.
3. The statistical significance of the expression levels was added to Figure 15a.
4. We feel sorry for our carelessness. In our resubmitted manuscript, the scientific name of the species and the name of the gene are written in italics.
5. We were sorry for our careless mistakes. In our resubmitted manuscript, the typo is revised. Thanks for your correction. In addition, We have rewritten lines 439, 460, and 496 according to the Reviewer’s suggestion.
6. We sincerely thank the reviewers for their careful reading. As suggested by the reviewers, we have corrected the strange symbols in lines 184, 185, 186, 198, 199, 200, and 324 to commas.
7. We have done our best to improve the manuscript and have made some changes to it. These changes do not affect the content or the framework of the thesis. And the spaces in lines 145, 146, 164, 186, 202, 290, 320, 365, 422, and 669 have been corrected.
8. The content of line 520 to line 525 has been removed.
9. In line 394, As suggested by the reviewer, we have corrected the “nutritional growth stage” into the “vegetative growth stage”.
We appreciate Editors/Reviewers’ warm work earnestly and hope the correction will meet with approval. Once again, thank you very much for your comments and suggestions.
Thank you for your consideration. I am looking forward to hearing from you.

Reviewer 3 Report
Please see attached.

Authors need to improve the writing style of this manuscript extensively. Also, should check for grammatical, spelling errors and the tense used.
Author Response
Dear Editors and Reviewers:
We feel great thanks for your professional review work on our article. As you are concerned, there are several problems that need to be addressed. According to your nice suggestions, we have made extensive corrections to our previous draft, the detailed corrections are listed below.
- Thanks for your suggestion. we do invite a friend of ours who is a native English speaker to help polish our article. And we hope the revised manuscript could be acceptable to you.
- We feel sorry for our carelessness. In our resubmitted manuscript, the scientific name of the species and the name of the gene are written in italics. In addition, the correct writing of the mutants and genes was checked throughout the manuscript.
- In lines 78, 79, 114, 162, 241,294,298, 296,329, 399, 451,485,490, 499,566, and 583, We have rewritten this part according to the Reviewer’s suggestion.
- In lines 237,438,448,460,524,634,676,740,810, we have removed certain inappropriate and repetitive content.
- We sincerely thank the reviewer for careful reading. As suggested by the reviewer, we have corrected the “salt mustard” into “Thellungiella”.
- We have reset the Figures to ensure better resolution.
- Line 231: TaRS genes are distributed in 16 chromosomes. The grouping of the three chromosome groups in wheat ABD in Figure 4b was a misunderstanding caused by our lack of clarity and precision and has now been corrected.
- We feel sorry for our carelessness. In our resubmitted manuscript, the typo is revised. Thanks for your correction.
- Line 294 & 298: we have corrected the “coarse goat amnion” into “Aegilops tauschii”.
- Figure 8b & 8c: The area of the circle represents the number of cis-acting elements.
- Figure 9: The purple ovals represent TaRS homologous proteins in Arabidopsis thaliana.
- In Figure 15a and Fig S6: added statistical significance and legend to the figure. Revise the title of Section 2.2.2 by the review comments.
- Section 2.2.2: As suggested by the reviewer, we have corrected the “recipient material” into “control”.
- We sincerely thank the reviewer for careful reading. As suggested by the reviewer, we have corrected the “Overexpression/ transgenic strain ” into “overexpression/transgenic line”.
- In the section beginning on line 555, a description of the drought test using soil is added before the description of the results begins.
- Line 557: Greening should be expressed as chlorosis. We apologize for any misunderstanding caused and have now corrected it.
- We were sorry for our careless mistakes. Thank you for your reminder. We have double-checked the section numbers.
- In Figure 17 and 18 legends, we have corrected the “PRO” into “proline”.
- We have improved the captions, and legends of subsection 2.2.4, Figure 19 and Figure 20.
- In Table 2, descriptions of the different letters have been added.
- We have made a simple improvement to Figure 21, but as the figure is intended to show trends in expression levels over time and due to the amount of data available, it can only be presented as a new submission, which we hope you will find satisfactory.
- We have made improvements to the methods section. The process of the generation of overexpression lines and related details have been added.
- In section 2.3.2. and 2.3.3, we compare the wild type, mock or empty vector and silenced lines for the stress assay, The addition of OE lines was intended to enrich the experiment on the one hand and to verify thoroughly that the tolerance of OE lines to drought and salt stress was derived from overexpression of the TaRS gene on the other.
We appreciate Editors/Reviewers’ warm work earnestly and hope the correction will meet with approval. Once again, thank you very much for your comments and suggestions.
Thank you for your consideration. I am looking forward to hearing from you.
